# Sound Change in Albanian Monolinguals and Albanian–English Sequential Bilingual Returnees in Tirana, Albania

Esther de Leeuw [1,*], Enkeleida Kapia [2,3] and Scott Lewis [1]

1 Department of Linguistics, School of Languages Linguistics and Film, Queen Mary University of London, London E1 4NS, UK; scott.lewis@qmul.ac.uk
2 Institute for Phonetics and Speech Processing, University of Munich, 80799 Munich, Germany; enkeleida.kapia@phonetik.uni-muenchen.de
3 Centre for Albanian Studies, Tirana, Albania
* Correspondence: e.deleeuw@qmul.ac.uk

**Abstract:** This research investigated contrastive perception of L1 phonological categories in Albanian–English bilinguals who returned to Albania after living abroad for over on average a decade. In Standard Albanian, there are phonemic contrasts between /c/ and /tʃ/, /ɫ/ and /l/, and /ɹ/ and /r/ . These phonemic contrasts do not occur in English. Using a "real speech" binary minimal pair identification task, we compared the accuracy and response times of bilingual returnees against functional Albanian monolinguals who had never lived abroad. Results showed that (1) reaction times for /c/ versus /tʃ/ were longest for both groups, indicating that this contrast was "harder" than the other contrasts. Surprisingly, (2) bilinguals outperformed monolinguals in accurately identifying /c/ versus /tʃ/; and (3) no significant group differences were found for the other two phonemic contrasts. In combination with other research showing that Albanian is undergoing a merger of /c/ and /tʃ/, our findings suggest that this merger is more advanced in monolinguals than bilinguals—probably because the bilinguals were abroad when the merger started. Examination of variation within the bilinguals indicated that (4) the younger the speaker was when they left Albania, and the more recently they had returned, the lower their accuracy was in identifying the laterals. These phonological findings enhance our understanding of perceptual L1 attrition whilst underlining the need to examine language change in the country of origin in L1 attrition research.

**Keywords:** sound change; bilingualism; perception; phonology; attrition; Albanian; English; returnees

## 1. Introduction

To date, first language (L1) attrition research has mainly focused on late bilinguals who live abroad. Such studies have found that within the domains of phonetics and phonology, long term pronunciation changes occur in the L1 of some, but not all, late bilinguals living in a second language (L2) environment. Very little is known about the native language (here, used synonymously with first language, i.e., L1) of bilinguals who return home after an extended stay in an L2 environment (but see Flores 2015, 2020, who has examined grammatical variables such as verb placement and gender marking). Although research into returnees is limited, it is generally proposed that late bilinguals who evidence L1 attrition abroad recover native language abilities upon returning to their home country (Major 1992; Yagmur et al. 1999). We were interested to find out whether this would be the case by examining perceptual abilities of late bilingual Albanian–English returnees who had lived in an Anglophone environment for on average over a decade.

Albanian is a novel language to examine within this context. Firstly, very little is known about the phonetics and phonology of Albanian, and the current study sheds light

on this dearth, particularly through examination of the monolingual group. Furthermore, Albania has a high rate of emigration. However, many highly educated Albanians have returned to Albania due to increased comparative prosperity in Albania (ILO 2014). This tendency to return to the country of origin aligns with other global movements as millions of people return to their country of origin every year, according to figures produced by the Organization for Economic Co-Operation and Development (OECD 2020). Therefore, the significance of our research is not only specific to Albanian returnees but rather to individuals all over the world who return to their home country after time abroad in an L2 environment.

More specifically, our study aimed to further the understanding of potential L1 restructuring in late bilingual returnees through examining both accuracy rates and response times on a minimal pair identification task in Albanian native speakers with English as a late L2. In codified Standard Albanian, the voiceless palatal plosive /c/ and the voiceless alveolar affricate /tʃ/ occur in contrastive distribution in words such as *qan* /can/ (Eng. 'cries') and *çan* /tʃan/ (Eng. 'breaks'); as do the lateral approximants /ɫ/ and /l/ in words such as *mall* /maɫ/ (Eng. 'nostalgia') and *mal* /mal/ (Eng. 'mountain'); and the voiced alveolar approximant /ɹ/ and voiced alveolar trill /r/ in words such as *rini* /ɹini/ (Eng. 'young') and *rrini* /rini/ (Eng. 'stay'), respectively (Camaj 1984; Dodi 2004; Memushaj 2011). Although some of these sounds exist in English (i.e., [l] and [ɫ] typically occur in complementary distribution in Standard British English, with [l] appearing in onset position, [ɫ] in coda; and both /tʃ/ and /ɹ/ occur in English (Wells 1982)), none of these phonemic contrasts found in Albanian occur in English. We, therefore, examined whether these contrasts would be perceived differently by the Albanian–English returnees than the functional Albanian monolinguals who had never lived abroad, potentially as a function of long-term English acquisition abroad (see, e.g., de Leeuw et al. 2018, who found L1 attrition within the phonological domain of production in Albanian–English bilinguals living in London, UK) and were also interested in variation within the bilingual returnees.

Indeed, a longitudinal study by Chang (2019) verified that adult native English speakers from the US who had recently arrived in Korea showed significant changes in their production of English plosives and vowels (in terms of voice onset time (VOT), fundamental frequency (*f*0), and formant frequencies) during Korean classes and that, crucially in relation to the current investigation, they continued to show altered English production a year later, months after their last Korean class. Interestingly, changes to their native language of English even persisted in some speakers who had returned to the US and reported limited active use of Korean in their daily life. Chang summarized that such findings indicate that the linguistic experience obtained in an L2 environment induces and prolongs restructuring of the L1 even after re-immersion in the language of the country of origin.

However, an earlier study by Sancier and Fowler (1997) showed that the VOT values of plosives in a female native speaker of Brazilian-Portuguese were shorter after a stay in Brazil of several months than after stays in the United States. It seemed that VOT values in both of her languages drifted towards her ambient language, in contrast to Chang's findings. As the speaker's Brazilian-Portuguese VOT values became shorter during her stay in Brazil, Sancier and Fowler interpreted that speakers are "disposed to imitate" (Sancier and Fowler 1997, p. 431). Specific to the current study, this would suggest that the perceptual abilities of the Albanian returnees would likewise return upon re-immersion in the L1 of Albanian, as similarly suggested by Major (1992) and Yagmur et al. (1999), unless prolonged changes occur after having lived abroad, as indicated by Chang (2019).

Both of the aforementioned studies, which investigated late bilinguals who returned to the L1 environment (albeit after a shorter stay than the current returnees), applied an acoustic analysis to measure pronunciation deviations in bilingual L1 speech in comparison to monolinguals. Such methodology, although rich in terms of speech production information, reveals less about the cognitive processes that underpin differences between monolingual and bilingual speech. Our study employed a "real speech" binary perception

task to investigate whether underlying representations in the L1 are modified in individuals who acquired English as a late L2 to a high degree of proficiency and, as such, expands on the aforementioned studies. It also builds upon previous studies by examining phonemic contrasts, rather than continuous phonetic variables, such as VOT, $f0$, and formant frequencies. As such, the results from this study have the potential to uncover the extent of long-term malleability in the perception of native speech phonemic categories.

### 1.1. Previous Research into Perceptual Identification in Late Bilinguals

Results from a small number of perception studies suggest that phonemic categories in the L1 are malleable in the case of late bilingualism. In general, these studies support the findings of investigations into production, which similarly indicate that L1 pronunciation changes when a new language is introduced late in life, both arising from short-term drift (e.g., Chang 2012; Sancier and Fowler 1997; Tobin et al. 2017) as well as long-term L1 attrition (de Leeuw 2019; de Leeuw et al. 2010, 2018; Dmitrieva et al. 2010; Hopp and Schmid 2013; Mayr et al. 2012).

In a recent study into perceptual cue-weighting, it was demonstrated that Russian-English bilinguals used different cue weighting depending on whether the language mode of investigation was Russian or English (Dmitrieva 2019). The bilinguals were all living in the United States at the time of recording, and the majority of them were described as first generation immigrants and late learners of English who were Russian-dominant at the time of participation (the majority of their day-by-day communication was conducted in Russian). These bilinguals nevertheless revealed cue weighting in an English mode that was not statistically different from that of the English monolinguals. Moreover, in Russian mode, they showed an increased reliance on vowel duration and a decreased reliance on glottal pulsing compared to monolingual Russians, indicating an influence from English. Cabrelli et al. (2019) similarly demonstrated an L2 influence on L1 perception in Brazilian-Portuguese (BP)–English bilinguals who were BP dominant and living in the United States through implementation of metalinguistic tasks, lower-level encoding tasks, and higher-level encoding tasks. Finally, Carlson (2018) examined phonetic processing of word initial sC clusters, which are illicit in Spanish but not in English, by late Spanish–English bilinguals, finding a weakened perceptual illusion of word-initial /e/ with a larger effect for learners immersed in L2 English. Together, these findings demonstrate that L2 experience affects perceptual patterns in bilinguals' L1 when bilinguals are immersed in their L2. Our study builds on these findings by examining bilinguals who have returned to their country of origin to see whether such long-term perceptual changes persist upon re-immersion in the L1.

Moreover, it has been demonstrated that novice L2 French learners show perceptual shifts in their L1 of English (Tice and Woodley 2012). Two tasks were implemented: a phoneme categorization task, in which participants identified CV syllables along a synthesized voice onset time (VOT) continuum (ba-pa, da-ta, and ga-ka), and a semantic priming task, in which it was observed whether more French-like VOT in tokens such as [kaet] instead of English-like VOT [k$^{h}$aet] would facilitate auditory semantic priming. During the first two weeks of the longitudinal study, which took place in the United States, i.e., with no L2 immersion, participants appeared to simply behave as native monolingual English speakers would; however, at weeks 3 and 4 of training, the participants indicated a shift in their English processing across both tasks, suggesting some amount of L1 category confusion as a result of their two weeks of L2 French input. At weeks 5 and 6, participants demonstrated a return to early behavior in the phoneme categorization task but not in the semantic priming task. The longitudinal findings from this study suggest that the introduction of an L2 into an individual's language repertoire actuates changes in the L1, even when there is no immersion in the L2 environment, i.e., with much less L2 input than Dmitrieva (2019), Cabrelli et al. (2019), and Carlson (2018).

However, in another recent study into word initial sC clusters in late Spanish–English bilinguals, for which data were collected both in the L1 and L2 environments, it was only

found that for the production task, but not for the perception task, bilinguals with more exposure to English and greater grammatical knowledge of English performed significantly more accurately than those with less exposure and lower grammatical knowledge (de Leeuw et al. 2019). Similarly, Parlato-Oliveira et al. (2010) found L1 resistance to L2 influence in phonological perception. Their study used an explicit vowel identification task and found that L1 Japanese–L2 Brazilian-Portuguese participants (who had been in the L2 environment of Brazil for 35 years) patterned with Japanese monolinguals. However, as Cabrelli et al. (2019) point out (see also de Leeuw 2019), this may have been because the participants had not yet actually fully acquired the relevant representation in the L2—although they were living in the L2 environment. In the current study, the late bilinguals had all acquired their L2 of English to a high degree of proficiency after living in an English environment for over a decade, so it is, therefore, possible that perceptual differences between the monolinguals and the bilinguals might be due to L2 acquisition of English.

Moreover, our participant cohort was quite different from the above studies (Cabrelli et al. 2019; Carlson et al. 2016; de Leeuw et al. 2019; Dmitrieva 2019; Parlato-Oliveira et al. 2010; Tice and Woodley 2012) because our bilinguals had all lived in the L2 environment and then returned to the L1 environment; thus, we are interested to find out whether prolonged perceptual changes persist in the L1 after re-immersion in the L1. However, it is also possible that any potential evidenced differences between the bilinguals and the monolinguals could be due to changes over time in the Albanian language in Albania (see Section "Voiceless palatal plosive /c/ and voiceless alveolar affricate /tʃ/ " and "Voiced alveolar approximant /ɹ/ and voiced alveolar trill /r/ " below).

Furthermore, our task examined three phonological contrasts within the L1 that do not have contrasting representations in the L2—the phonemic contrasts simply do not occur in English. We were, therefore, interested to observe whether, if restructuring occurred at all, there would be differences in accuracy rates and RTs between the three phonemic contrasts. Finally, in contrast to the aforementioned studies, we were also interested to see whether we would be able to observe variation within the bilingual returnees, for example, as a result of (i) their age when they left Albania; (ii) the amount of time that they had lived abroad; (iii) the amount of time they had been living in Albania after their return; and (iv) their self-assessed amount of English they continued to still speak in Albania (%Eng).

*1.2. Albanian Language*

Albanian, known as *shqip* in Albanian, is a language of the Indo-European family with 6–7 million speakers who live mostly in the Republic of Albania and Republic of Kosovo but also in Italy, Greece, Macedonia, and Montenegro. It is widely accepted that Albanian forms a branch of its own within the Indo-European language family (e.g., Bopp 1855; Pedersen 1897; Çabej 1976). Even though its origin has been debated, it is generally accepted that Albanian originates from Illyrian (Hetzer 1995).

Modern Albanian has two main dialects: Gheg (Geg in Albanian), spoken in Central and Northern Albania, and Tosk (Tosk in Albanian), spoken in the South of the country. Compared to the two major dialects of Gheg and Tosk, Standard Albanian is relatively new; it was first established in 1972 at the National Congress of Orthography in Tirana as a result of the socio-political decision to unify the country under the umbrella of one language. In fact, the motto of the Congress of Orthography was the idea that the nation should have one literary language (Kostallari 1973, 1984). At the time, the idea was to suppress dialectical preferences in the interest of important national needs. Even though the Congress decided that the basis for the standard variety would be the Tosk dialect (an idea that had been popular since 19th century Albanology), scholars have recognized and mostly agreed that the Standard was not just a copy of the Tosk dialect, but contained many features of Gheg, especially in morphosyntax (Kostallari 1973; Domi 1973; Hetzer 1995; Memushaj 1996, 2013; Ismajli 2005; but see, e.g., Pipa 1989 for another point of view). While before the creation of the standard variety people exercised writing in literary variants of both Tosk and Gheg, after 1972, it became compulsory for everyone to use the Standard, be

it at the workplace or within the school system. Dialects were disallowed in public media and public forums but continued to be used locally and within families.

The fall of communism in Albania in 1989 marked a drastic change in the language practices of Albanians. Many Albanians left the country in search of better opportunities. However, as has been reported, many have recently opted to return for a variety of reasons (Cena and Heim 2022). It is the speech of such returnees from English-speaking countries that is the focus of this study.

*1.3. Relevant Consonant Pairs*

1.3.1. Voiceless Palatal Plosive /c/ and Voiceless Alveolar Affricate /tʃ/

Both the voiceless palatal plosive /c/ and the voiceless alveolar affricate /tʃ/ occur in onset, medial, and coda positions in minimal pairs in Albanian (Lowman 1932; Camaj 1984; Beci 2004). Orthographically, /c/ is represented as <q> and /tʃ/ is represented as <ç>. Beci (2004) and other traditional phonetic manuscripts in Albanology (i.e., Dodi 2004; Memushaj 2011) consider /c/ to be a palatal plosive, as transcribed here (but see Lowman (1932) and Coretta et al. (2021), who argue that /c/ is an alveo-palatal affricate). The phoneme /tʃ/ is a relatively common sound in the languages of the world, and, as in these languages, in Albanian, also, it is often described as a palato-alveolar affricate (Lowman 1932; Beci 2004). Importantly, neither of these two phonemes has been mentioned as being more functionally important than the other in the grammar or phonology of Albanian, and both are considered to not be common sounds (Jubani-Bengu 2005).

Some dialectology studies have noted that in the area of Northeastern and Northwestern Gheg, /c/ in final word position is often articulated as [tʃ], e.g., <kaq> 'this much' (Eng. food bite as [katʃ] 'bite' rather than [kac] (Gjinari et al. 2007; Shkurtaj 2013) which is often attributed to contact that these areas had with Serbian (Ajeti 1978; Agani 1978). Recently, there have been impressionistic and empirical reports, however, that both these sounds are merging in other Gheg varieties and in Tosk (Kolgjini 2004; Periskopi 2017). The reasons for this recent merger are not entirely clear, but some researchers have suggested that the renewed prestige of Gheg after the strict language policies during communism fell could be one explanation (Kolgjini 2004). During communism, the Gheg variety was considered to be less prestigious than Standard Albanian and potentially less prestigious than Tosk, but since the end of communism in 1989, Gheg has regained its prestige, even within the Albanian-speaking literatti of Kosovo and Albania (Constantine 2016).

For our study, we were interested in considering this potential sound change (see the recent study by Coretta et al. (2021), who specify that <q> is actually an affricate). It is often proposed that sound changes are led by younger female speakers (Milroy and Milroy 1985; Labov 1990; Williams and Kerswill 1999; Fridland 2008; Harrington et al. 2008; Lewis et al. 2019) so in examining our data, we considered it relevant to examine whether this sound change might have been led by younger females as well. Therefore, we examined interpersonal variation within both the monolingual and bilingual groups to see whether younger females might actually be "worse" at identifying these sounds than e.g., older groups.

1.3.2. Lateral Approximants

A few studies have looked at the contrastive distribution between light and dark lateral approximants in modern Albanian and have concluded that light /l/ is an alveolar lateral, while dark /ɫ/ is an alveo-dental lateral; both occur in onset, medial, and coda position in minimal pairs (Lowman 1932; Camaj 1984; Beci 2004). Orthographically, light /l/ is represented as <l>, and dark /ɫ/ is represented as <ll>. Dodi (1970) and Beci (2004) have shown that F2 frequency is comparatively higher in the light lateral and lower in the dark lateral. Specifically, they provide the following values (gender was not specified), provided in Table 1 below.

**Table 1.** F1 and F2 values for /l/ and /ɬ/.

|  | F1 | | F2 | |
|---|---|---|---|---|
|  | **Dodi (1970)** | **Beci (2004)** | **Dodi (1970)** | **Beci (2004)** |
| /l/ | 198 Hz | 211 Hz | 1550 Hz | 1383 Hz |
| /ɬ/ | 254 Hz | 275 Hz | 950 Hz | 927 Hz |

Moreover, this contrastive distribution is robust, with neither of these phonemes mentioned as more functionally important than the other in the grammar or phonology of the language. This robustness was illustrated by findings from perception studies in both adults and children, with Albanian as an L1, in which adults were able to perceptually differentiate between the light and dark phonemes in pseudo-words (Müller 2015), and children were able to perceptually differentiate the two sounds even at 3 years of age (the study did not look at children younger than 3) in real words (Müller and Kapia 2019). However, in a study examining Albanian native speakers in London, UK, it was found that this contrast is vulnerable to L1 attrition; in particular, /l/ in coda position became darker, evidencing lower F2 values (de Leeuw et al. 2018).

### 1.3.3. Voiced Alveolar Approximant /ɹ/ and Voiced Alveolar Trill /r/

The Albanian rhotics /ɹ/ (voiced alveolar approximant) and /r/ (voiced alveolar trill) occur in onset, medial, and coda position in minimal pairs (Lowman 1932; Camaj 1984; Beci 2004). Orthographically, /ɹ/ is represented as <r>, and /r/ is represented as <rr>. Studies on the /ɹ/ and /r/ contrast are quite limited, but according to the only official language atlas produced in Albanology, this contrast exists in almost all variants of both dialects, except for a pocket in the northeastern variety of Tosk (Gjinari et al. 2007). Importantly, however, it was suggested that the rhotic contrast might have already begun to disappear from a few Tosk varieties, not just the northeastern one, even when the Congress of Orthography took place in 1972 and that the decision for the contrast to be kept in the standard was to allow differentiation of some word pairs with high functional load (Ismajli 2005). Later impressionistic studies and some case study analyses have brought forth evidence in favor of the disappearance thesis (e.g., Jubani 2012; Hysenaj 2009; Belluscio 2016), suggesting that /r/ might currently be undergoing a shift towards the direction of /ɹ/ in a widespread manner in different varieties of Albanian.

### 1.3.4. Objectives

The primary objective of this research was to examine the extent to which minimal pairs in the native language (i.e., L1) of late bilingual returnees were identified after prolonged immersion in the L2 (which does not feature those same minimal pairs). To investigate this objective, we administered a minimal pair contrastive perception task to a group of Albanian–English bilingual returnees and a group of functional Albanian monolinguals who had never lived abroad.

We assumed variation within the bilinguals, so our second objective was to investigate factors that might have influenced the bilinguals' accuracy and response times. The reason we thought that there could perhaps be some Albanian–English bilinguals who evidenced prolonged perceptual L1 attrition is that de Leeuw et al. (2018) showed attrition at the level of production in the contrast of light <l> versus dark /ɬ/ in Albanian L1 speakers living in the UK. Could these findings be extended to the level of perception in a novel group of returnees? More generally, some evidence suggests that multilingual listeners exhibit language-specific patterns in perception, or 'language modes' (Antoniou et al. 2012; Gonzales and Lotto 2013; Grosjean 2001). Moreover, first language speech perception can diverge from the monolingual norms due to the effects of additional languages (Chang 2016; Celata and Cancila 2010; Dmitrieva et al. 2020). We, therefore, looked at the variables of (i) age when bilinguals left Albania (ALA); (ii) amount of years that they had lived abroad (YLA); (iii) amount of years they had been living in Albania after their return (YAR);

and (iv) self-assessed amount of English they continued to still speak in Albania (%Eng). Moreover, given the dialectal variation described above, we examined whether dialectal background would influence perception in both the bilinguals and monolinguals.

Three phonological contrasts which exist in Albanian, but not in English, were tested and both accuracy rates and response times were compared: /c/ versus /tʃ/; /ɫ/ versus /l/; and /ɹ/ versus /r/. Our third objective was therefore to see whether there would be differences in accuracy and response times between the different minimal pairs in both the bilingual and monolingual groups.

Indeed, we considered it plausible for there to be differences in the way these sound contrasts would be identified. For the rhotics and the affricate–plosive contrast, there is a scenario in which the L2 has only one of the sounds. For the laterals, there is a contrastive distinction in the L1 surfacing as an allophonic alternation in the L2, at least in British English and some varieties of North American English (Wells 1982). L2 exposure for the lateral contrast would immerse the bilingual in a setting in which the laterals are not contrastive and, with extensive exposure, this could, in turn, affect how this contrast is perceived in the L1. This would not happen for the other two contrasts, for which an L2 speaker could well spend years and years in an immersion setting without hearing one of the members of the contrast (i.e., /r/, /c/) in L2 speech; therefore, there would be no competition to the function of this contrast but rather simply a lack of input of one of the sounds in each contrast.

Ours was an extreme "real language" test of perceptual attrition in returnees. We wanted to see whether this binary identification task—was the word correctly identified or not?—could reveal long-lasting perceptual attrition effects in this group of returnees.

## 2. Materials and Methods

Data collection for this experiment took place in Tirana, Albania, in a quiet room at the Academy of Albanological Studies in May 2018. Both the first and second author were present in the room when the data were collected, but the actual experiment was delivered in Albanian on the computer.

### 2.1. Participants

Data from 30 participants were collected and each participant filled in an adapted version of the MPI Language Background Questionnaire (Gullberg and Indefrey 2003), which had been translated into Albanian and took approximately 15 min to complete (see Appendix A). One person who we initially considered to be monolingual was excluded because he had lived in Turkey for an extended amount of time.

Of the remaining 29 participants, 18 were considered to be monolingual because they predominantly spoke Albanian in their daily lives and had never lived abroad, and 11 were considered to be bilingual returnees because they had lived abroad in an English environment for an extended period of time (minimum of 3 years, maximum of 18 years) and then returned to Albania (see Table 2). Although all the functional monolinguals also knew other languages, including English, none had ever been immersed in these languages. Notably, many of the bilingual returnees, in addition to the monolinguals, reported knowledge of other languages, such as Italian, but had not lived in an Italian speaking environment and did not continue to speak Italian in their regular lives. Some of the bilinguals continued to speak English in their daily lives in Albania, for example, because they worked in an English language environment, and their self-assessment of how much English they continued to speak was also documented through the questionnaire (see Table 2). All of the returnees had moved abroad after early childhood with the earliest move being at 11 years of age and the latest move at 28 years of age.

**Table 2.** Participant background summary information.

| | N | Female: Male Ratio | Age at Time of Experiment | Age Moved Abroad | Years Lived Abroad | Years Ago Returned | % English Spoken Daily |
|---|---|---|---|---|---|---|---|
| Albanian monolinguals | 18 | 11:7 | 25.6 (6.5) Min: 17 Max: 40 | n/a | n/a | n/a | 9.2 Min: 0 Max: 50 |
| Albanian–English bilingual returnees | 11 | 6:5 | 37.5 (7.6) Min: 23 Max: 45 | 18.5 (5.1) Min: 11 Max: 28.5 | 11.5 (4.8) Min: 3 Max: 13 | 7.5 (3.5) Min: 1 Max: 18 | 37.2 (24.6) Min: 0 Max: 80 |

*2.2. Stimuli Recordings*

The sound files, which the participants listened to, were real words containing the phonemes in question (see Appendix B for a full list of words with IPA transcriptions and English translations). These words were pre-recorded in Tirana at the Academy of Albanological Studies. The speaker was a 54-year-old professional speaker of Standard Albanian who had taught Standard Albanian for over 30 years and worked as a radio broadcaster earlier in her career and spoke Standard Albanian.. She was known at the National Institute of Linguistics to speak a so-called "clean" Standard Albanian, and we considered her to, therefore, be ideal for this task as it was crucial that she would indeed articulate the phonemic contrasts, and furthermore, that other dialectal features would not influence the listener's perceptions of her articulations. We wanted to have a real speaker produce the words, rather than, e.g., computer-generated speech, because we considered any potential findings to be more meaningful if they were founded in the perception of real human speech, with all of its complexities, rather than in computer-generated speech.

The speaker, who was unaware of the purpose of the experiment, was instructed to read the words presented to her on a printed sheet of paper one by one, at a normal speed rate, and in Standard Albanian. The second author recorded the speaker using Speech Recorder (Draxler and Jänsch 2004) and a Beyerdynamic Custom One Pro Headset. Once the words had been recorded, they were isolated and saved as individual sound files, which were thereafter implemented in the minimal pair perception task, designed in PsychoPy by the first author (Peirce and MacAskill 2018).

*2.3. Experimental Procedure*

Participants entered the quiet room where the outline of the procedure was described to them in Albanian and then filled in the participant consent form, which was also in Albanian.

Thereafter, the language background questionnaire was filled in, as described above. Once this was completed, the actual experimental procedure was explained to the participant, and when the participant had no other questions, they put on the headphones and started the minimal pair identification task on the computer in front of them.

As already stated, the purpose of the phoneme identification task was to determine whether there would be a difference between bilingual returnees and monolinguals in accuracy and response times in the minimal pair identification task that focused on (1) /l/ versus /ɫ/; (2) /c/ versus /tʃ/ and (3) /ɹ/ versus /r/. Participants heard a word that contained one of the contrasts in question, e.g., they heard the word *lum* [lum] (Eng. 'river') and, on the screen, they saw both *lum* and *llum* and were asked to choose which word they had heard. In this case, the correct answer would have been *lum*, because that is the word they heard.

Specifically, participants were requested to press the left arrow key if the word they heard was on the left-hand side of the screen, whereas if the word they had heard was on the right-hand side of the screen, they should press the right arrow key. The words and trial number appeared together on the screen at the start of each trial so participants would know how far through the experiment they had progressed. After 500 ms (during which they could read the two words on the screen), the recording was played (which

contained one of the two words presented on the screen). The participants had 3 s after the onset of the recording to determine whether the word they had heard over their headset was on the left or right of the screen. Within this time, they had to make their decision and press either arrow key, at which point the next slide occurred. If they made their decision earlier, the next trial would progress automatically. Therefore, in total, each identification trial lasted a maximum of 3.5 s. A total of 25 minimal pairs were played to the participants (see Appendix B for full list of minimal pairs), and each token was repeated twice, totaling 100 target tokens (see Table 3). In addition to the minimal pairs, there were 16 distractor minimal pairs, which were also repeated twice, so in total, there were 64 distractor tokens. Therefore, in total, 164 audio recordings were played to the participants in the main experiment. There was no break during this session and words were automatically randomized in PsychoPy for each trial to avoid tiredness effects.

**Table 3.** Summary of tested contrasts.

|  | Contrast | Example Contrast | Nr. of Minimal Pairs | Nr. of Tokens |
|---|---|---|---|---|
| Condition 1: lateral contrast | /l/ versus /ɬ/ | *lum* versus *llum* | 10 | 40 |
| Condition 2: plosive/affricate contrast | /c/ versus /tʃ/ | *qan* versus *çan* | 7 | 28 |
| Condition 3: rhotic contrast | /ɹ/ versus /r/ | *rini* versus *rrini* | 8 | 32 |

Preceding the main experimental presentation, there was initially a practice session in which four tokens were played with word presentation in alternating order: *po–jo* (Eng. 'yes'–'no') and *bor–dor* (Eng. 'snow'–'hand'), repeated twice (see Figure 1). Including the onscreen explanation, which was delivered in Albanian, the practice session took approximately 1–2 min.

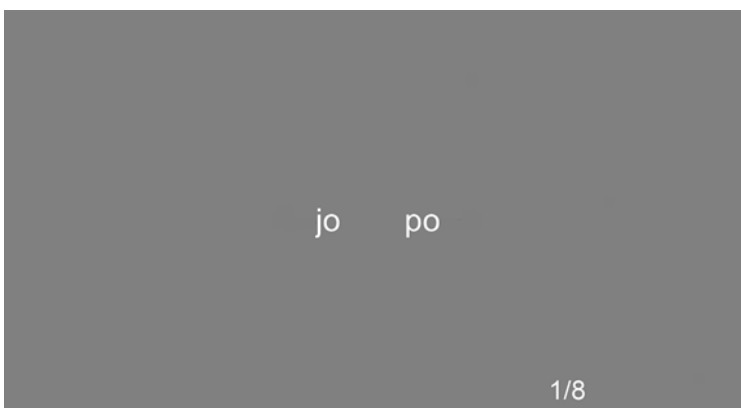

**Figure 1.** Example from the practice session of the identification task. Participants heard a single word over headphones and chose whether the word they heard was on the left or right by pressing the left or right arrow key.

The maximum time allocated for the entire experiment was 10 min; however most participants finished in 8–9 min because the screen progressed automatically once they had made their decision.

### 2.4. Statistical Analysis

For both the perception and production results, data were organized in CSV files using Excel software. Thereafter, *R* (R Core Team 2019) was used for the analyses and a series of binomial mixed-effects regression models were built for the accuracy results of the perception task, and a series of linear regression models were built for the reaction time results, using the *lme4* (Bates et al. 2015) and *lmerTest* (Kuznetsova et al. 2017) packages in *R*, respectively. These packages allowed the influence of fixed and random factors on response

variables to be examined. For the analysis of the perception data, a total of 2900 responses were collected. Timed out responses (i.e., trials that were not identified within 3500 ms) were marked as incorrect (7/2900). In terms of response times, all incorrect responses were excluded (161/2900), yielding a total of 2739 responses. We present the accuracy results initially, then the response time results.

## 3. Results

### 3.1. Monolinguals versus Bilingual Returnees

3.1.1. Accuracy

In order to assess the extent to which minimal pairs in the native language of Albanian sequential bilingual returnees were successfully identified, it was first necessary to examine the accuracy results from the minimal pair contrastive perception task, focusing on the potential differences in accuracy rates between monolingual and bilingual speakers.

To that end, a mixed effects logistic regression was conducted in *R* (R Core Team 2019) using the *glmer* function of the *lme4* package (Bates et al. 2015). The modeled response was accuracy (i.e., *yes* or *no* as to whether the participant could accurately identify the correct phoneme). Fixed effects were incorporated for mon versus bi (monolingualism versus bilingualism), condition (1; laterals versus 2; plosives/affricates versus 3; rhotics) and dialect (Gheg or Tosk). An interaction term was also included for mon versus bi and condition. This was so that any potential differences between monolingual and bilingual participants in identifying particular minimal pair contrasts could be ascertained. Random intercepts were included for participant, token, and position (onset versus medial versus coda), and by-participant random slopes were incorporated for condition. A likelihood ratio test was performed to compare the model that contained the interaction term against one that did not. The model with the interaction performed significantly better than the one without it ($\chi^2(2) = 10.416$, $p < 0.001$).

As the fitted model contained a factor with multiple grouping levels, an analysis of deviance was conducted using the *Anova* function of the *Car* package (Fox and Weisberg 2018). This provided an ANOVA-like overview of results (Table 4). Mon versus bi was not found to significantly impact accuracy rates ($p = 0.872$). Dialect was found to be significant ($\chi^2(2) = 4.945$, $p = 0.026$), with Gheg speakers displaying higher accuracy rates than Tosk speakers (*z-ratio* = 2.224, $p = 0.026$), as determined through a post-hoc pairwise comparison conducted using *emmeans* (Lenth 2019). Condition was also found to be significant ($\chi^2(2) = 22.930$, $p < 0.001$), as was the interaction term ($\chi^2(2) = 12.310$, $p = 0.002$). Given condition's involvement in a significant interaction, only the interaction is considered further.

**Table 4.** Analysis of deviance table (Type II Wald chi-square tests) for the fixed effects tested in the accuracy experiment. The model includes monolinguals and bilingual returnees.

|  | $X^2$ | *Df* | *p* |
|---|---|---|---|
| mon versus bi | 0.026 | 1 | 0.872 |
| Condition | 22.930 | 2 | <0.001 |
| Dialect | 4.945 | 1 | 0.026 |
| mon versus bi × condition | 12.310 | 2 | 0.002 |

To explore the interaction between mon versus bi and condition, post-hoc pairwise comparisons were conducted using the *emmeans* package (Lenth 2019). As can be observed in Table 5, monolinguals differed significantly from the bilingual returnees in accurately identifying plosives and affricates (*z-ratio* = −2.940, $p = 0.003$). Interestingly, bilingual returnees outperformed monolinguals in accurately identifying /c/ from /tʃ/. They did not differ significantly for rhotics (*z-ratio* = 1.751, $p = 0.080$) nor laterals (*z-ratio* = 1.037, $p = 0.299$), although the difference for rhotics approached significance. Accuracy rates were

generally high at around 95%, except for the monolingual /c/ versus /ʧ/ identification, which was just above 85% (see Figure 2).

**Table 5.** Pairwise comparisons of accuracy rates between monolinguals and bilingual returnees for each condition. 'Estimate' shows the difference in estimated marginal means on the log odds ratio scale. Results are averaged over the levels of dialect.

| Contrast | Condition | Estimate | SE | z-Ratio | *p* |
|---|---|---|---|---|---|
| Monolinguals– Bilingual returnees | Laterals | 0.466 | 0.450 | 1.037 | 0.299 |
| | Plosive and affricate | −1.317 | 0.448 | −2.940 | 0.003 |
| | Rhotics | 0.732 | 0.418 | 1.751 | 0.080 |

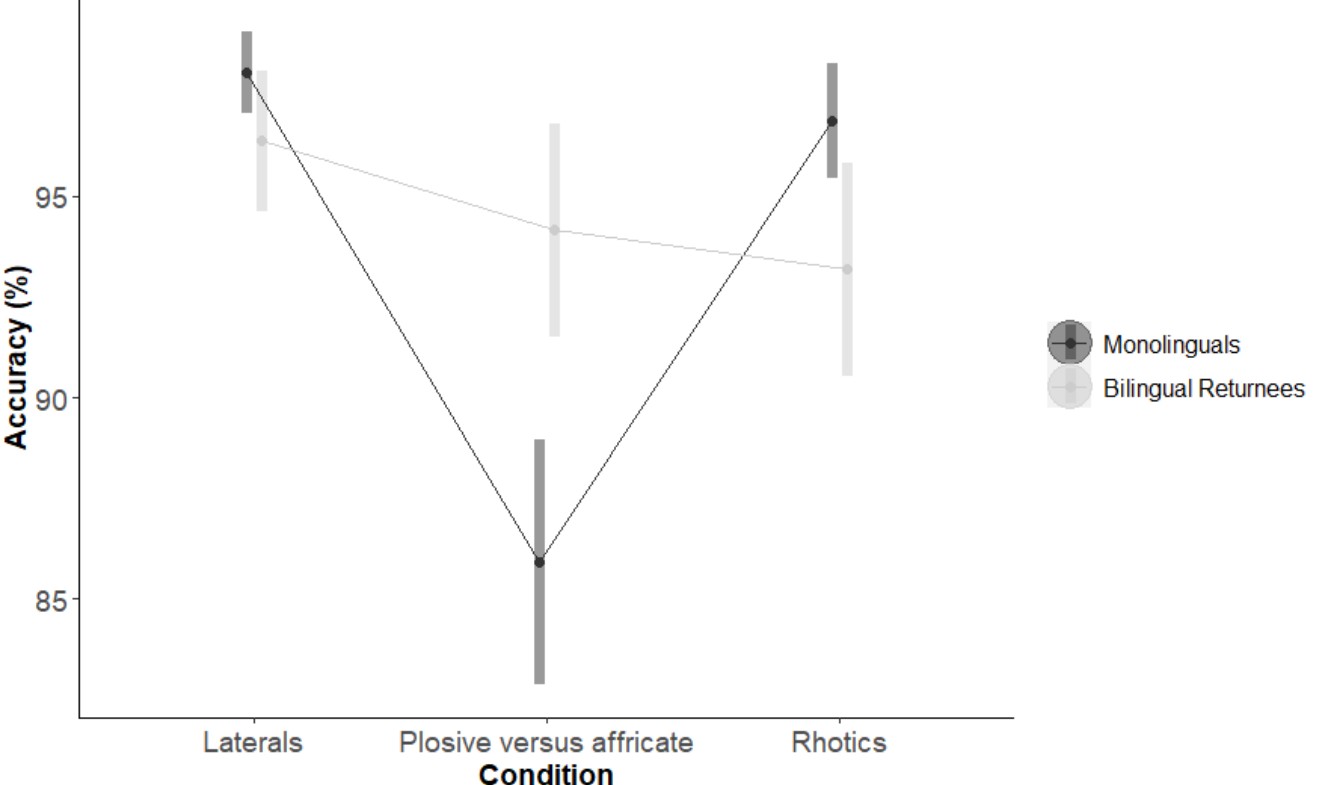

**Figure 2.** Perception accuracy in Albanian–English bilingual returnees versus Albanian monolingual speakers across each condition. The vertical bars represent 95% confidence intervals.

To see whether outliers were potentially enhancing, or even causing these significant differences, we plotted boxplots with outliers for the monolinguals and bilinguals for each contrast.

With regard to /c/ versus /ʧ/, the monolingual subject 24 (68% accuracy) was an outlier (see Figure 3) from Burrel, a town in northern Albania wherein the Gheg variety is not expected to contain this contrast, at least in word final position. However, both monolingual subjects 26 (54% accuracy) and 29 (71% accuracy) were born and grew up in Tirana, where this contrast is considered to be active. Moreover, the bilingual subject 19, who moved to the United States when he was 11 years old and returned to Tirana three years before the data were recorded, grew up in Vlora, an area in which Tosk is spoken, and this contrast is functional; therefore, his low accuracy rate of 75% is notable.

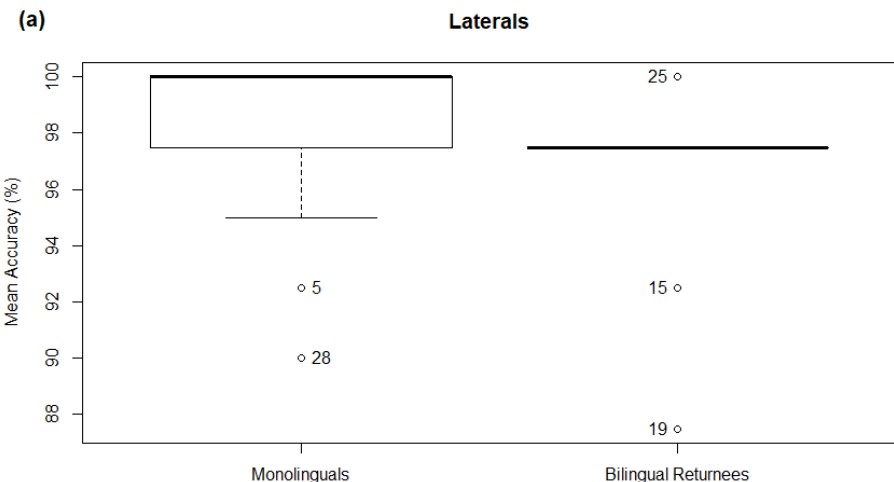

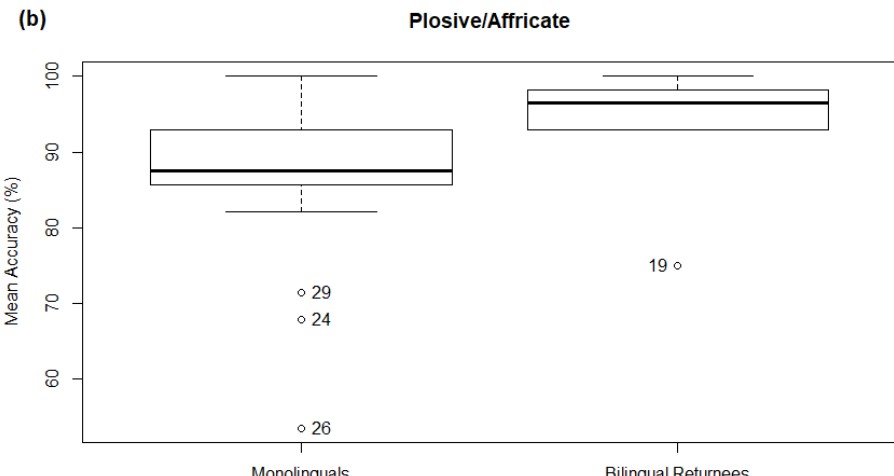

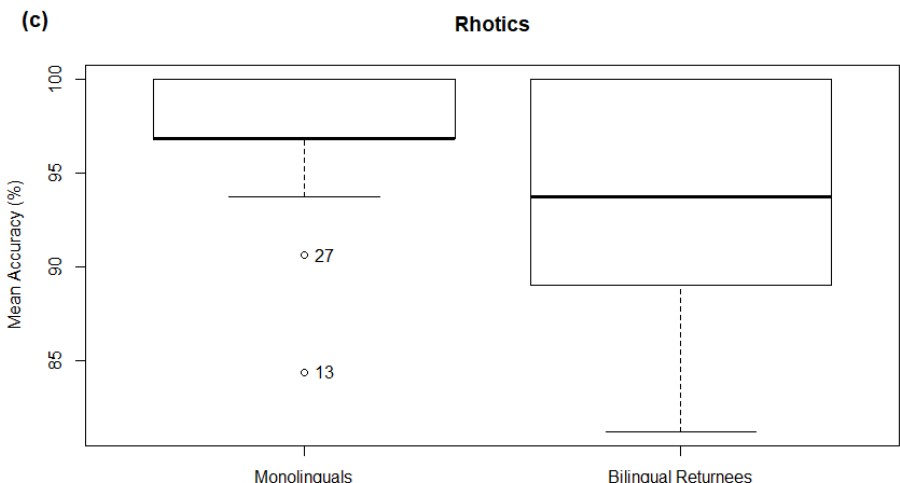

**Figure 3.** Boxplots depicting accuracy rates for each of the three contrasts: (**a**) shows percentage accuracy in the identification of /l/ versus /ɬ/; (**b**) /c/ versus /tʃ/; (**c**) /ɹ/ versus /r/. In each panel, monolinguals and bilingual returnees are shown separately. Outlier participants are also displayed in each panel.

To see whether the higher accuracy rates in the bilinguals than the monolinguals could be due to a sound change in Albanian—which the bilinguals did not undergo to the same extent as the monolinguals because the bilinguals were not resident in Albania—we conducted Pearson's correlation tests in order to observe whether younger speakers, regardless of dialect, were more likely to have lower accuracy rates on the minimal pair identification task. As can be observed in Figure 4, we indeed found that there was a positive correlation between age and accuracy ($\rho = 0.534$, $p = 0.003$) with younger speakers performing less accurately than older speakers, although there was no correlation between age and response time. Looking at male and female speakers separately (Figure 5), the correlation was significant only for females ($\rho = 0.619$, $p = 0.008$), which is in line with findings from other studies that indicate that sound changes are typically led by female speakers (Milroy and Milroy 1985; Labov 1990; Williams and Kerswill 1999; Fridland 2008; Harrington et al. 2008; Lewis et al. 2019).

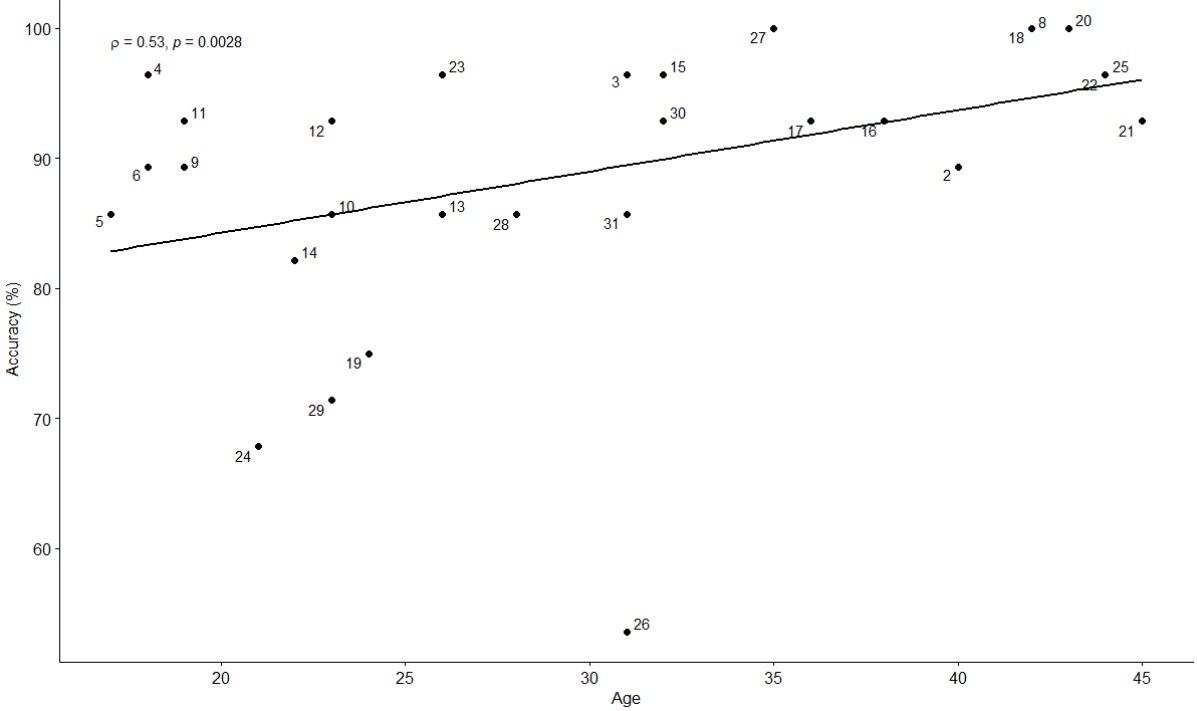

**Figure 4.** Scatterplot displaying the relationship between age and percentage accuracy in the identification of the contrast between /c/ and /tʃ/. Spearman's correlation coefficient is indicated by ρ. Corresponding *p*-values are found to the right of each coefficient value.

Figure 6 illustrates that, in terms of raw percentages, /tʃ/ was more accurately identified than /c/ by both the monolingual and bilingual groups. If this were true, it could be taken to lend support to either a merger of /tʃ/ and /c/ > /tʃ/ and /or /tʃ/ in the L2 of English of the bilinguals enhancing the identification of these sounds, at least in the bilinguals. What the raw data appear to show is that for both groups, /c/ seems to be "harder" to identify than /tʃ/.

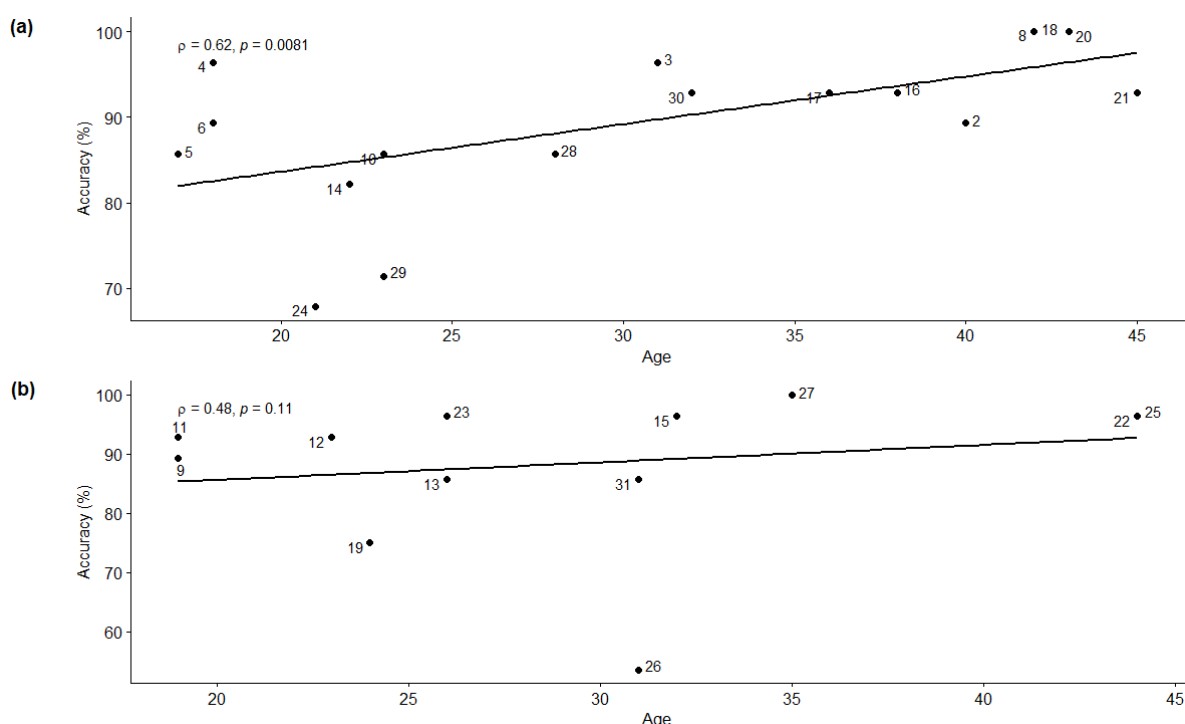

**Figure 5.** Scatterplots displaying the relationship between age and percentage accuracy in the identification of the contrast between /c/ and /ʧ/: (**a**) depicts results for females; (**b**) shows the results for males. Spearman's correlation coefficient is indicated by ρ. Corresponding *p*-values are found to the right of each coefficient value.

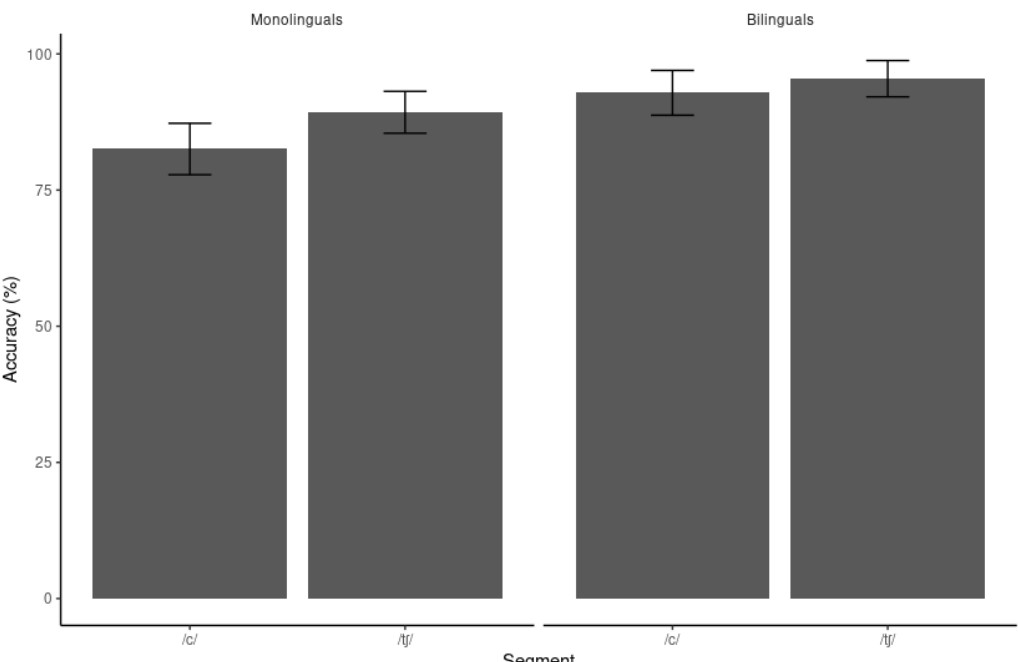

**Figure 6.** Perception accuracy in Albanian–English bilingual returnees and Albanian monolingual speakers for the plosive–affricate contrast. 95% confidence intervals are included.

We sought to test whether this observation was supported by statistical data. To that end, we conducted a mixed effects logistic regression, taking accuracy to be the modeled response. Fixed effects were included for mon versus bi and segment (/c/ versus /ʧ/). Random intercepts were included for participant and token. By participant random slopes

were incorporated for segment. The inclusion of an interaction term for the fixed effects did not improve model fit and was, therefore, excluded from the final model. However, in this case, segment was not found to significantly impact accuracy rates ($\chi^2(1) = 1.0665$, $p = 0.302$).

With regard to /l/ versus /ɬ/, the monolingual subjects 5 (93%) and 28 (90%) had accuracy levels lower than the monolingual mean (see Figure 3). For the bilingual returnees, subject 15 had a lower accuracy level than the bilingual mean (93%). The bilingual returnee subject 19 had the lowest accuracy score at 88% for /l/ versus /ɬ/.

With regard to the rhotics (see Figure 3), there was overall more variation within the bilinguals than the monolinguals. The monolingual subject 13 (84%) was from an area within the Northeastern variety of Tosk where the /r/ versus /ɹ/ contrast is not present; however, subject 27 (91%) was from an area where this distinction is supposed to be active. Interestingly, although the bilingual returnee subject 19 evidenced the lowest accuracy rates for the lateral and plosive versus affricate contrasts, for the rhotic contrast, his accuracy rate was 100%.

### 3.1.2. Reaction Times

A linear mixed effects model was created for the reaction time data using the *lmerTest* package (Kuznetsova et al. 2017). The modeled response was log-transformed reaction time. Fixed effects were included for mon versus bi (monolingual versus bilingual), condition and dialect. An interaction term was included for mon versus bi and condition. Random intercepts were included for token, position, and participant, and by-participant random slopes were used for condition. To obtain $p$ values for main effects and the interaction term, a type III analysis of variance table was produced for the fitted model by calling the *anova* function. The results, summarized in Table 6 below, indicate that only condition ($F(2, 55) = 17.783$, $p < 0.001$) and dialect ($F(1, 26) = 4.837$, $p = 0.037$) were significant, although reaction time differences between monolinguals and bilingual returnees approached significance ($F(1, 27) = 3.106$, $p = 0.090$). The interaction term was not significant ($F(2, 27) = 1.210$, $p = 0.314$).

**Table 6.** Type III Analysis of variance table for the fixed effects tested in the reaction time experiment. The model includes monolinguals and bilingual returnees.

|  | Sum Sq | Mean Sq | NumDF | DenDF | *F* Value | Pr (>*F*) |
|---|---|---|---|---|---|---|
| mon versus bi | 0.112 | 0.112 | 1 | 26.547 | 3.106 | 0.090 |
| condition | 1.28 | 0.641 | 2 | 55.406 | 17.783 | <0.001 |
| dialect | 0.174 | 0.174 | 1 | 26.034 | 4.837 | 0.037 |
| mon versus bi × condition | 0.087 | 0.044 | 2 | 26.737 | 1.210 | 0.314 |

In terms of dialect, Gheg speakers were found to display longer reaction times than Tosk speakers. To look at reaction time differences as a function of condition, pairwise comparisons with Bonferroni adjusted *p*-values were conducted in *emmeans* (see Table 7). The difference between laterals and plosives and affricates was found to be significant (*t-ratio* = −5.737, *p* < 0.001), indicating that the lateral contrast was identified significantly more quickly than the plosive versus affricate contrast. Similarly, laterals and rhotics were found to differ (*t-ratio* = −3.610, *p* = 0.002) with the contrast for laterals eliciting a significantly faster reaction time than the rhotic contrast. Finally, response times for the plosive versus affricate contrast and rhotic contrast were also found to differ significantly (*t-ratio* = 3.199, *p* = 0.007) with reaction times for rhotics being shorter. Therefore, reaction times were longest for the plosive/affricate contrast, intermediate for rhotics, and shortest for laterals, although no significant group differences between monolinguals and bilingual returnees were found (see Table 8).

**Table 7.** Pairwise comparisons of reaction time by condition. 'Estimate' shows the difference in estimated marginal means on the log scale. Results are averaged over the levels of mono/bilingualism, dialect, and sex.

| Contrast | Estimate | SE | DF | t-Ratio | *p* |
|---|---|---|---|---|---|
| Laterals–Plosive and affricate | −0.212 | 0.037 | 59.4 | −5.737 | <0.001 |
| Laterals–Rhotics | −0.114 | 0.031 | 49.7 | −3.610 | 0.002 |
| Plosive and affricate–Rhotics | 0.099 | 0.031 | 59.3 | 3.1999 | 0.007 |

**Table 8.** Mean reaction times (ms) and standard deviations for each of the tested contrasts for monolinguals and bilinguals.

| Contrast | Group | Mean Reaction Time (ms) | Standard Deviation |
|---|---|---|---|
| /l/ versus /ɫ/ | Monolinguals | 989 | 277 |
| | Bilinguals | 1047 | 295 |
| /c/ versus /ʧ/ | Monolinguals | 1118 | 346 |
| | Bilinguals | 1210 | 363 |
| /ɹ/ versus /r/ | Monolinguals | 1021 | 263 |
| | Bilinguals | 1120 | 320 |

*3.2. Bilingual Returnees*

3.2.1. Accuracy

The secondary objective of this research was to investigate potential factors that could have impacted bilingual returnees' accuracy rates and reaction times. As previously mentioned, these factors were (1) the speaker's age when he/she left Albania (ALA); (2) the duration of time, in years, that the speaker had lived in an English-speaking country outside of Albania (YLA); (3) the number of years since they had returned to Albania at the time of the experiment (YAR); and (4) the amount of English they continued to speak in Albania (%Eng). Variables were considered individually in relation to each of the three minimal pair contrasts of interest (i.e., laterals, plosive versus affricate, and rhotics). Attempts were made to explore each of these variables in relation to condition in a generalized linear mixed effects model. However, plotting multiple interaction terms resulted in model overfitting. As we were interested in the interaction between each of the variables and condition, we turned instead to correlation tests. To establish the relationship between each factor and bilinguals' accuracy rates, it was first necessary to ascertain mean percentage accuracy scores for each condition for each bilingual returnee. As mean percentage scores did not uniformly show normal distributions, non-parametric, Spearman's rank correlation tests were conducted using the *cor.test* function of the *stats* package.

The correlation results for laterals, summarized in Figure 7 below, revealed a positive correlation between ALA (age when left Albania) and percentage accuracy ($\rho = 0.715$, $p = 0.013$), indicating that the older the speaker was at the time that they left Albania, the greater their accuracy in identifying the contrast between laterals. A positive correlation was also found for YAR (years since returned to Albania) and percentage accuracy ($\rho = 0.678$, $p = 0.022$), signifying that the longer ago a participant had returned to Albania, the better their accuracy in identifying /l/ from /ɫ/; however, both of these effects were largely driven by subject 19, who left Albania when he was 11 and returned three years earlier, and subject 15, who left when he was 12 and had also only returned three years ago. Re-running these correlation tests without participants 15 and 19 removed the significant effects for both ALA and YAR (both *p*s > 0.05).

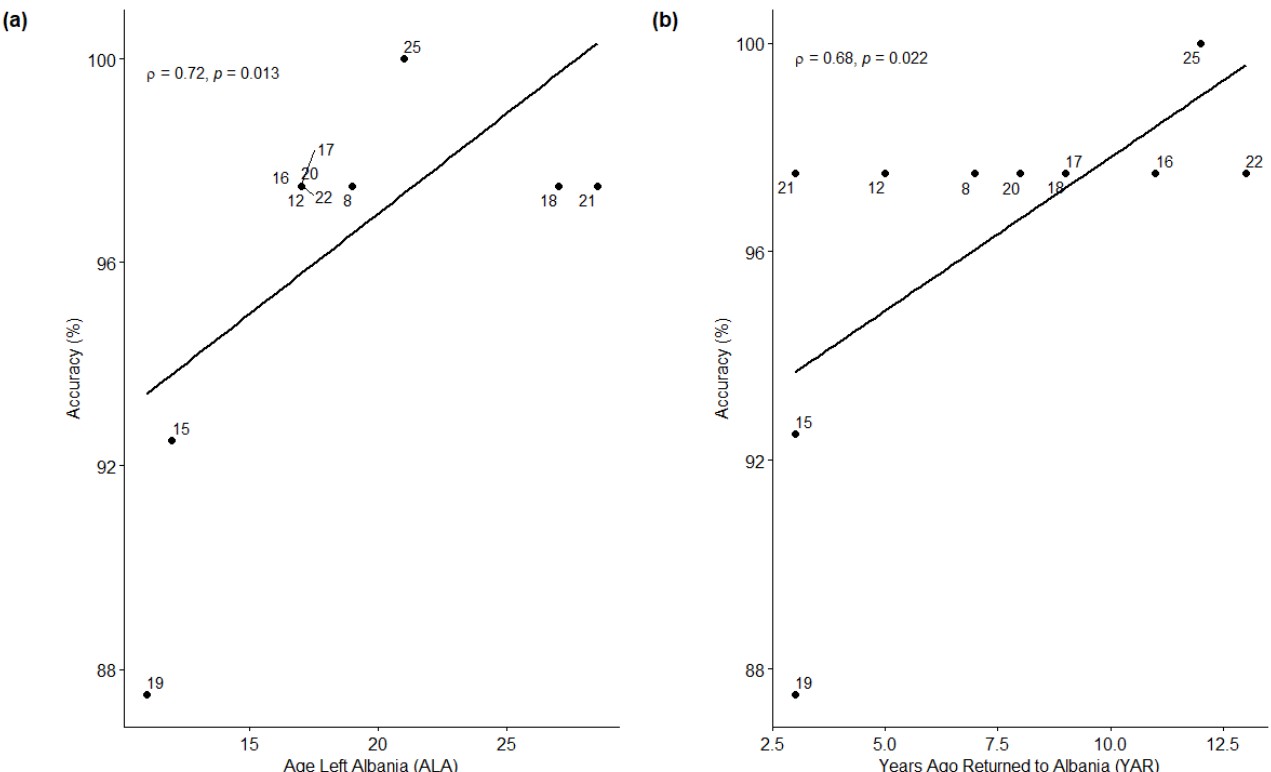

**Figure 7.** Scatterplots displaying the relationship between (**a**) ALA; and (**b**) YAR and percentage accuracy in the identification of laterals. Spearman's correlation coefficient is indicated by ρ. Corresponding *p*-values are found to the right of each coefficient value.

Neither YLA nor %Eng were found to significantly correlate with perceptual accuracy for laterals, and no significant correlations were found between any of the tested conditions and accuracy in identifying the contrast between /ɹ/ and /r/, nor between /c/ and /tʃ/.

### 3.2.2. Reaction Times

Although the distinction between reaction times for monolinguals versus bilingual returnees was not significant, we nevertheless decided to explore factors that may have impacted reaction times for the bilingual returnees because the result approached significance. We included the same factors as in the accuracy analysis: (1) ALA, (2) YLA, (3) YAR, and (4) %Eng. These factors were included as fixed effects in a linear mixed effects regression, alongside condition. Interaction terms were included for ALA and condition, YLA and condition, YAR and condition, as well as %Eng and condition. Log-transformed reaction time functioned as the modeled response and random intercepts were included for token, position, and participant. By participant, random slopes were incorporated for condition. No fixed effects or interaction terms were found to be significant.

### 4. Discussion

The primary objective of this research was to examine the extent to which minimal pairs of Albanian were identified by late Albanian–English bilingual returnees who had resided in an Anglophone country for, on average, over a decade. This binary task was an extreme "real language" test of perceptual attrition: is this naturally produced word correctly identified or not? There were no gradual changes between the phonemic pairs as we really wanted to get to the crux of perceptual attrition and see whether the Albanian words were correctly identified. The results indicated that there was no significant difference between the monolinguals and bilingual returnees in accurately identifying minimal pairs containing /l/ versus /ɫ/, nor those containing /r/ versus /ɹ/ (although the latter contrast almost reached significance with bilinguals performing on average less accurately

than monolinguals). Interestingly, however, the bilinguals outperformed the monolinguals in correctly identifying /c/ and /ʧ/, as discussed in more detail shortly.

Before entering into the discussion in more detail, we emphasize that we do not know whether the bilingual returnees in our study had indeed undergone perceptual L1 attrition in their L2 environment, because our study was not longitudinal, i.e., data were not collected in the Anglophone countries in which they had previously resided. Therefore, we cannot be sure whether, for example, the bilinguals' ability to identify the rhotics and laterals was weaker in the Anglophone environment or, for the sake of argument, whether the ability to identify the plosive from affricate might have been even better in the L2 environment. This piece of information is missing in our research, and future endeavors into returnees would ideally collect data at least twice before and after the move. Nevertheless, in their own right, we consider our findings to be interesting because they examine a novel group of bilinguals: Albanian L1–English L2 returnees living in Albania.

Furthermore, we cannot be sure whether our results are transferable to larger populations because we have small participant numbers. With larger participant numbers, we may have found stronger, or weaker effects. Future studies examining perceptual L1 attrition should ideally recruit more participants; however, some populations are challenging to recruit, and this does not mean that they should not be studied or the results from their small numbers not reported. Indeed, in the current study, which took place in Albania, the poorest European country, there were challenges in collecting data. For example, a number of power outages occurred during data collection causing data to be lost and/or the collection to be started anew; there was no properly sound-proofed room available to collect data, so collection at times needed to be restarted during noisy environments (e.g., thunder storms and/or construction site nearby); and transportation was not reliable, causing changes in schedules and even dropouts. In more developed countries, these challenges occur to a lesser extent; yet, we believe that gathering data from unique populations, such as the current returnees, is worthwhile to enrich our overall understanding of language and to spawn new research. It is with this in mind that we interpret our findings.

Now to the actual discussion. Although group comparisons showed no significant difference for /l/ versus /ɫ/ nor for /r/ versus /ɹ/ (although the latter approached significance), with regard to the second objective, there was evidence that some bilingual returnees did evidence persistent perceptual L1 attrition. This was especially notable for subject 19 who only had an 88% accuracy rate on the lateral contrast, the lowest accuracy rate of all participants for this phonemic contrast. This returnee had been living in Albania for three years since his return; however, he was the only participant to consider English his native language and Albanian his L2 in the questionnaire, even though he grew up in a completely Albanian environment until he was 11 years of age, when he moved to the United States (the youngest age of departure from Albania of all bilinguals).

As previously mentioned, in terms of the distinction between /c/ and /ʧ/, the bilingual returnees outperformed the monolinguals in accurately identifying these phonemes. On the one hand, this finding aligns with literature which has revealed perceptual restructuring of the L1 in the L2 environment (Cabrelli et al. 2019; Carlson et al. 2016; de Leeuw et al. 2019; Dmitrieva 2019; Parlato-Oliveira et al. 2010; Tice and Woodley 2012) because a significant difference was observed between the monolinguals and the returnees. However, our finding is somewhat unique as the bilinguals not only differed from the monolinguals, they actually outperformed the monolinguals. It may be that, because this contrast appeared to be the "hardest" of the three contrasts (i.e., accuracy rates were lowest for this contrast in both the monolinguals and the bilinguals, and response times for both groups were significantly longer in comparison to the two other contrasts), acquiring a new language might actually have facilitated the identification of these two phonemes. For example, although from another language domain, it was found that the orthographic representations of the native language change with L2 experience, which boosts markedness perception in the L1 (Duñabeitia et al. 2020). It could be that these effects generalize to

the phonological domain, and that phonemic perception in the L1 is also boosted upon L2 acquisition, at least for certain phonemic contrasts.

Within this contrast, only the phoneme /ʧ/ is present in English, and it could be that not hearing (or indeed, hearing it less frequently, as they were immersed in an L2 English environment with limited exposure to Albanian) the phoneme /c/ for, in some cases, decades could have heightened the bilinguals' capacity to identify /c/, rather than impaired it. Notably, the bilinguals showed more accuracy in identifying /c/ than the monolinguals did. However, with this argument, that English L2 exposure would have heightened the bilinguals' identification of Albanian phonemic contrasts, we should likewise have expected the bilinguals to outperform the monolinguals on the rhotic contrast, as only /ɹ/ is present in English. In fact, we found, if anything, the opposite, that the bilinguals on average scored lower than the monolinguals, although this difference was not significant. Therefore, for our findings, we tend not to promote the interpretation that L2 exposure promoted the bilinguals' enhanced performance on this contrast but rather that something else was at play.

Moreover, it is again notable that although, on average, the bilinguals outperformed the monolinguals on identifying /c/ and /ʧ/, subject 19 again scored only 75% on this identification task (n.b. he grew up in the town of Vlora where Tosk is spoken and this contrast is traditionally considered to be productive), so this interpretation, that L2 exposure improves perception in the L1, is not generalizable to the other contrasts, nor is it generalizable to all participants.

Another explanation for the finding that bilingual returnees outperformed monolinguals on the /c/ versus /ʧ/ identification task may be dialectal differences, because this contrast is not considered to be fully present in Gheg dialects (Gjinari et al. 2007; Shkurtaj 2013), which is potentially caused by contact with Serbian, a language that does not have this contrast (Ajeti 1978; Agani 1978). However, only the monolingual subject 24, who scored 68% on this identification task, was from Burrel, a town where a Gheg variety is spoken in which this contrast is not considered to be fully productive. Both monolingual subjects 26 and 29, who also had lower accuracy scores, were born and grew up in cities where this contrast is considered to be productive (Tirana). Therefore, it does not appear that dialectal differences would adequately explain why the bilinguals outperformed the monolinguals on this phonemic contrast.

Another explanation—which we believe to be most likely—is that there is a sound change in progress (merging of /c/ and /ʧ/ to /ʧ/) occurring in Albanian that the monolinguals have participated in but the bilinguals have not—at least to the same extent—because for a part of this sound change, they were living abroad. Future research would need to investigate this possibility by exploring different age groups in different areas and towns in Albania to see whether younger speakers were more likely to evidence the merging of these sounds than older speakers, which would point towards a sound change in progress. We found that, in terms of raw percentages, both groups of speakers were more accurate at identifying /ʧ/ than /c/, supporting the proposition that there is a merger to /ʧ/ (Gjinari et al. 2007; Shkurtaj 2013). In this instance, however, this point was not reinforced by the statistical findings, potentially due to our relatively small number of participants.

Nevertheless, younger females displayed the lowest accuracy levels, suggesting that they are leading a change in which the phonemic contrast between /c/ and /ʧ/ is being lost. This is in line with Kolgjini (2004), who showed that a merger is occurring between the two sounds that began in the Gheg variety and is now spreading to Tosk. She notes that the merging process was halted due to the strict political, communist language policies during the communist era. Now that the political constraints of these language policies have disintegrated, the merger is not just limited to a few Gheg varieties (Kolgjini 2004).

Our finding that the merging of these sounds is being led by younger females in both groups is in line with a large body of research suggesting that sound changes are often led by young females (Milroy and Milroy 1985; Labov 1990; Williams and Kerswill 1999; Fridland 2008; Harrington et al. 2008; Lewis et al. 2019). For our purposes, we considered it

meaningful that the bilinguals outperformed the monolinguals on this phonemic contrast, although one bilingual scored relatively low (again, participant 19, 75%, who was also the youngest bilingual participant).

As we observed differences between bilingual returnees, such as the lower accuracy rates of subject 19, our second objective was to investigate factors that might have influenced the bilinguals' accuracy and response times, such as the (i) age when they left Albania (ALA); (ii) years that they had lived abroad (YLA); (iii) amount of time they had been living in Albania after their return (YAR); and (iv) self-assessed amount of English they continued to speak in Albania (%Eng). Only two correlation tests were found to be significant with regard to accuracy rates in the bilinguals, and these were both for the lateral contrast. The younger the speaker was when he or she left Albania, the lower their accuracy in identifying the contrast between laterals. This might be because phonemic contrasts are more susceptible to attrition in sequential bilinguals who learn their L2 at a younger age when those same sounds are in complementary distribution in the competing L2. However, it is noteworthy that in de Leeuw et al. (2018), the age of L2 acquisition was not correlated with increased phonological attrition of the /l/ versus /ɫ/ contrast. Furthermore, the more recently a participant had returned to Albania, the lower his or her ability to correctly identify the lateral contrast, potentially because speakers had less time to re-adjust to the phonological patterns of their ambient environment. Both of these effects were largely driven by subjects 15 and 19, who had both left Albania in early adolescence and returned three years before the data for this study were collected. It seems that particularly participants 15 and 19 may have had more lasting perceptual attritional effects that continued after their return to Albania.

Nonetheless, with regard to the third objective, it is noteworthy that these same correlations were not found for the other phonemic contrasts. What is unique about the /l/ versus /ɫ/ contrast in comparison to the other examined phonemic contrasts? Firstly, both of these phonemes are approximants, and—although they are indeed functional phonemes in Albanian—acoustically, the differences between them are gradient; whilst the manners of articulation differ more categorically between /c/ (plosive) and /tʃ/ (affricate) and /r/ (trill) and /ɹ/ (approximant). It could be that the more gradient differences between /l/ and /ɫ/ were harder to perceptually re-acquire on the part of the returnees and that, likewise, they are more susceptible to attrition in younger bilinguals., i.e., the lateral contrast is simply less stable than the other two contrasts. In a larger sample group, with a more wide-spread of ALA and YAR, these correlations for the lateral may be stronger and likewise, a significant difference between monolinguals and bilingual returnees may arise, as shown in de Leeuw et al. (2018) for speech production of this phonemic contrast in Albanian.

For both monolinguals and bilingual returnees, reaction times were longest for the /c/ (plosive) versus /tʃ/ (affricate) contrast, followed by the /r/ (trill) versus /ɹ/ (approximant) and then shortest for the /l/ versus /ɫ/ contrast. This finding indicates that some phonemic contrasts are simply harder to identify than others, regardless of whether an individual is monolingual or bilingual. Similarly, it is particularly interesting to note that subject 19, who reported that he still used English 80% of the time in his daily life in Tirana, accurately identified /r/ from /ɹ/ 100% of the time, but /l/ from /ɫ/ 88% and /c/ from /tʃ/ 75%, again suggesting that certain contrastive identification tasks are "harder" than others.

In general, our findings suggest that bilingual returnees adapt to the ambient language, as suggested by Major (1992), Sancier and Fowler (1997), and Yagmur et al. (1999), because there were no significant group differences between /l/ versus /ɫ/, nor between /r/ versus /ɹ/. However, this interpretation is not conclusive because we did find both group and individual differences, e.g., the bilingual returnees had significantly higher accuracy rates on the /c/ versus /tʃ/ contrast; and bilingual returnees who had moved abroad at a younger age and who had more recently returned to Albania had significantly lower accuracy rates on /l/ versus /ɫ/ identification.

Had all contrasts delivered similar results (e.g., for all contrasts monolinguals had outperformed bilinguals or there had been no significant differences between monolinguals

and bilinguals for any of the contrasts), this could have been considered to be evidence for a systematic perceptual shift in the L1 (Chang 2012; Guion 2003; Mayr et al. 2012). However, we did not find this. Instead, there were differences between the contrasts: no significant group differences between monolinguals and returnees for the /l/ versus /ɫ/ contrast, nor for the /r/ versus /ɹ/ contrast; however, bilinguals outperformed monolinguals for the /c/ versus /ʧ/ contrast. Therefore, if perceptual L1 attrition occurred at all (as evidenced by, e.g., participant 19), we consider that these differences between phonemic contrasts are supportive of counterpart-by-counterpart prolonged restructuring of the L1 (see, e.g., de Leeuw 2019; Sankoff 2004) rather than a systematic shift in the L1.

It is important to emphasize that although our returnees were re-immersed in the Albanian language—due to the fact that they lived in Albania—many continued to speak English in their daily lives. For example, participant 19, who showed the most perceptual restructuring in his Albanian, reported that he used English 80% of the time in Tirana. It may, therefore, be possible that any effects of L2 English on the L1 Albanian were not due to attritional processes whilst living in the L2 environment but rather that restructuring took place after his return to Albania, or potentially, in combination, and is ongoing. If a merger of /c/ and /ʧ/ is taking place in Albanian, it may be that he is undergoing this merger (note his 75% accuracy rate on this contrast) and that he has undergone perceptual L1 attrition that persists after re-immersion in Albania. Indeed, there is some research that suggests that at least in speech production, language markers fulfill socio-indexical functions on the part of the heritage language speakers in their L2 (Alam and Stuart-Smith 2011; Heselwood and McChrystal 1999; Kirkham 2011; Sharma and Sankaran 2011), as well as potentially affecting their L1 (Lewis et al. 2019; Passoni et al. 2018; Nodari et al. 2019). If such social indexing evidenced in speech production in turn affects speech perception, it may be that the bilinguals, upon returning to their country of origin, delineate themselves from the monolingual Albanians through changes in their L1, which affect the perceptual domain.

In addition to our findings regarding malleability of the L1 perceptual system, i.e., our lateral findings showing interpersonal variation (subjects 15 and 19), as well as our contribution to language change in Albanian, our study makes important contributions with regards to Albanian phonetics and phonology. For example, our results suggest that these three contrasts, particularly the /l/ versus /ɫ/ and the /r/ versus /ɹ/ contrast, are robust and functionally important in Albanian, given the overall high accuracy rates of the monolinguals. They also suggest that even in the monolinguals, the /c/ versus /ʧ/ contrast was more difficult than the rhotic and lateral contrasts, because these latter two contrasts had higher accuracy rates and shorter response times. Further perceptual research may reveal dialectal differences in active sound changes in process occurring in Albanian (e.g., Jubani 2012; Hysenaj 2009; Belluscio 2016).

Moreover, previous research has shown that the lateral contrast in Albanian does undergo phonological attrition in late bilinguals at the level of speech production when bilinguals are in the L2 environment (de Leeuw et al. 2018). Therefore, if L2 speech production and perception are intrinsically connected, as postulated by most models in L2 speech acquisition (see, e.g., PAM, Best 1995, LP, Escudero 2009; SLM-r, Flege and Bohn 2021), the findings of the present research tend to support the understanding that returnees adapt their speech perception to the ambient language (Major 1992; Sancier and Fowler 1997; Yagmur et al. 1999). This suggests that any perceptual L1 attrition that might have occurred in the L2 environment—if it occurred at all—did not persist upon re-immersion in Albanian.

In general, the findings from this study—an identification task with naturally produced stimuli (to be entered in the analyses as correct versus incorrect responses)—tell us that the phonological domain is not impacted consistently in dual language settings (because not all contrasts were impacted in the same way), and that exposure to an L2, even after re-immersion in the L1 environment, affects phonemic contrasts varyingly and individuals differently. Had our test been more sensitive (i.e., incorporating manipulated gradual changes between the phonemic contrasts), we might have been able to observe more

nuanced differences between the two groups, and this would be encouraged in future research. Nevertheless, we believe that our findings regarding the malleability of speech perception are important because they show both perceptual changes in individual bilingual returnees (subjects 15 and 19) and in the monolinguals who are most likely undergoing a merging of the /c/ and /tʃ/ contrast, with the bilinguals lagging behind in this merger.

**Author Contributions:** The research idea was conceptualized by E.d.L. with feedback on the Albanian language from E.K. The experiment was programmed in PsychoPy by E.d.L. The recordings of the Standard Albanian speaker were made by E.K. in Tirana. The data were collected in Tirana and then analysed by E.d.L., E.K. and S.L. The statistics in R were conducted by S.L. with feedback from E.d.L. The manuscript was written together with E.d.L. focusing on the introduction, methodology and discussion, S.L. focusing on the results and E.K. focusing on the Albanian language. All authors have read and agreed to the published version of the manuscript.

**Funding:** This research was funded by the British Academy grant SRG18R1\180902 "Permanency of L1 Attrition of Multiple Linguistic Domains in Returning Albanian Migrants from the UK and USA" which was awarded to the first (PI) and second author (co-PI). The research was also supported by an Alexander von Humboldt Alumni Fellowship to the first author called "I Don't Hear a Difference": Perceptual L1 Attrition in Albanian and an Alexander von Humboldt Foundation Fellowship awarded to the second author. The second author was also supported by the ERC grant 742289 titled 'Interaccent' awarded to Professor Jonathan Harrington by the European Union's Horizon 2020 research and innovation programme.

**Institutional Review Board Statement:** "I Don't Hear a Difference": Perceptual L1 Attrition in Albanian" was granted ethical approval from Queen Mary University of London on 24 April 2018 (QMREC1791). The research proposal was considered to be extremely low risk; and did not require the scrutiny of the full Research Ethics Committee.

**Informed Consent Statement:** In line with ethical approval, informed consent was obtained from all subjects involved in the study.

**Data Availability Statement:** Further data can be made available by contacting the first author.

**Acknowledgments:** We would like to thank the Institute of Linguistics and Literature in Tirana for allowing us to use their premises to conduct the experiment. We are grateful to Anila Omari who was the speaker for our recordings. We would also like to thank Gentian Zyberi, International Law and Human Rights at the Norwegian Centre for Human Rights, for helping us to recruit participants in Tirana, and all of the participants who offered their valuable time to take part in our study.

**Conflicts of Interest:** The authors declare no conflict of interest. The funders had no role in the design of the study; in the collection, analyses, or interpretation of data; in the writing of the manuscript, or in the decision to publish the results.

## Appendix A. Pyetësori mbi të Dhënat Gjuhësore

Ky pyetësor është përshtatur nga Pyetësori mbi të dhënat gjuhësore hartuar në Institutin e Psikolinguistikës Max Planck, në departamentin e Dinamikës së Procesimit Multilingual në Nijmegen, nga M. Gullberg dhe P. Indefrey (2003)

Studiuesi: Nr. i Informatorit: Inicialet e Informatorit: Data:

**A.**

1. Cilën dorë përdorni: 2. Shikimi: 3. Mosha: 4. Gjinia:

5. Cila është diploma juaj më e lartë (shkollë e mesme, universitet, master, etj):

6. Çfarë profesioni keni (p.sh., student, jurist):

7. Ku keni lindur? ☐ Shqipëri ☐ Vend tjetër_______________________

8. Në cilin qytet keni lindur? _______________________

9. Për sa kohë keni jetuar në qytetin e lindjes? _______________________

10. Nëse keni jetuar në ndonjë qytet/vend tjetër, ku keni jetuar dhe për sa kohë keni qënë aty?

_______________________________________________________________________________________

_______________________________________________________________________________________

11. Ku jetoni momentalisht?          ☐ Shqipëri          ☐ Vend tjetër_________________

12. Ku jeni rritur? _____________________________________________________________________

13.  A do e quanit veten folës të një dialekti të caktuar të shqipes? Nëse po, cilin dialekt flisni?

_______________________________________________________________________________________

**B.**

14. Cila është gjuha juaj amtare?          ☐ Shqipja          ☐Gjuhë tjetër_________________

15. A ka ndonjë gjuhë tjetër që do ta quanit gjuhën tuaj amtare? Nëse po, cilën?

_______________________________________________________________________________________

16. Listoni të gjitha gjuhët e tjera që flisni, duke i listuar nga ajo që flisni më mirë tek ajo që nuk e flisni dhe aq mirë.

| Gjuha |
|:-----:|
| 1 |
| 2 |
| 3 |
| 4 |
| 5 |
| 6 |

**C.**

17.  Për të gjitha gjuhët që listuat më lart, thoni sa mirë i përdorni sipas shkallës së më-poshtme:

Jo mirë          1          2          3          4          5          Shumë mirë

| Gjuha | Të folurit | Të kuptuarit | Të shkruarit | Të lexuarit | Gramatika | Shqiptimi |
|:-----:|:----------:|:------------:|:------------:|:-----------:|:---------:|:---------:|
| 1 | | | | | | |
| 2 | | | | | | |
| 3 | | | | | | |
| 4 | | | | | | |
| 5 | | | | | | |
| 6 | | | | | | |

18. Për secilën gjuhë që keni përmendur më lart, përcaktoni sa shpesh e përdorni mesatar-isht (në mënyrë active) këtë gjuhe gjatë një jave. Sigurohuni që përqindjet që do të raportoni të formojnë shumën 100%.

| Never = 0% | Half the time=50% | All of the time=100% |
|------------|-------------------|----------------------|
| Kurrrë = 0% | Gjysmën e kohës = 50% | Gjithë kohën = 100% |

| Language Gjuha | 0% | 10% | 20% | 30% | 40% | 50% | 60% | 70% | 80% | 90% | 100% |
|---|---|---|---|---|---|---|---|---|---|---|---|
| **1** | | | | | | | | | | | |
| **2** | | | | | | | | | | | |
| **3** | | | | | | | | | | | |
| **4** | | | | | | | | | | | |
| **5** | | | | | | | | | | | |
| **6** | | | | | | | | | | | |

19. Nëse keni ndonjë koment tjetër që ju duket me rëndësi për mënyrën se si i përdorni këto gjuhë, shkuajeni këtu, ju lutem:

_____________________________________________________________________________________
_____________________________________________________________________________________

20. Komentet e studiuesit

**Appendix B**

| | |
|---|---|
| mjel | mjell |
| mjell | mjel |
| mjel | mjell |
| mjell | mjel |
| vjel | vjell |
| vjell | vjel |
| vjel | vjell |
| vjell | vjel |
| thel | thell |
| thell | thel |
| thel | thell |
| thell | thel |
| gjel | gjell |
| gjell | gjel |
| gjel | gjell |
| gjell | gjel |
| tul | tull |
| tull | tul |
| tul | tull |
| tull | tul |
| lum | llum |
| llum | lum |
| lum | llum |
| llum | lum |
| loj | lloj |
| lloj | loj |
| loj | lloj |
| lloj | loj |
| lak | llak |
| llak | lak |
| lak | llak |
| llak | lak |
| mal | mall |
| mall | mal |
| mal | mall |
| mall | mal |
| djal | djall |
| djall | djal |
| djal | djall |

| | |
|---|---|
| djall | djal |
| quk | çuk |
| çuk | quk |
| quk | çuk |
| çuk | quk |
| qorri | çorri |
| çorri | qorri |
| qorri | çorri |
| çorri | qorri |
| fuqi | fuçi |
| fuçi | fuqi |
| fuqi | fuçi |
| fuçi | fuqi |
| poqa | poça |
| poça | poqa |
| poqa | poça |
| poça | poqa |
| qan | çan |
| çan | qan |
| qan | çan |
| çan | qan |
| qarë | çarë |
| çarë | qarë |
| qarë | çarë |
| çarë | qarë |
| qaj | çaj |
| çaj | qaj |
| qaj | çaj |
| çaj | qaj |
| rite | rrite |
| rrite | rite |
| rite | rrite |
| rrite | rite |
| resht | rresht |
| rresht | resht |
| resht | rresht |
| rresht | resht |
| deri | derri |
| derri | deri |
| deri | derri |
| derri | deri |
| ruaj | rruaj |
| rruaj | ruaj |
| ruaj | rruaj |
| rruaj | ruaj |
| ura | urra |
| urra | ura |
| ura | urra |
| urra | ura |
| kor | korr |
| korr | kor |
| kor | korr |
| korr | kor |
| roje | rroje |
| rroje | roje |

| | |
|---|---|
| roje | rroje |
| rroje | roje |
| rasë | rrasë |
| rrasë | rasë |
| rasë | rrasë |
| rrasë | rasë |
| bor | dor |
| dor | bor |
| bor | dor |
| dor | bor |
| po | jo |
| jo | po |
| po | jo |
| jo | po |
| tok | kok |
| kok | tok |
| tok | kok |
| kok | tok |
| far | bar |
| bar | far |
| far | bar |
| bar | far |
| mes | ves |
| ves | mes |
| mes | ves |
| ves | mes |
| fis | pis |
| pis | fis |
| fis | pis |
| pis | fis |
| grua | krua |
| krua | grua |
| grua | krua |
| krua | grua |
| burr | furr |
| furr | burr |
| burr | furr |
| furr | burr |
| bor | dor |
| dor | bor |
| bor | dor |
| dor | bor |
| po | jo |
| jo | po |
| po | jo |
| jo | po |
| tok | kok |
| kok | tok |
| tok | kok |
| kok | tok |
| far | bar |
| bar | far |
| far | bar |
| bar | far |
| mes | ves |
| ves | mes |
| mes | ves |

| | |
|---|---|
| ves | mes |
| fis | pis |
| pis | fis |
| fis | pis |
| pis | fis |
| grua | krua |
| krua | grua |
| grua | krua |
| krua | grua |
| burr | furr |
| furr | burr |
| burr | furr |
| furr | burr |

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
