# Peer review of "Sound Change in Albanian Monolinguals and Albanian–English Sequential Bilingual Returnees in Tirana, Albania"

_languages, doi:10.3390/languages8010080_

Round 1

Reviewer 1 Report

The study “Sound Change in Albanian Monolinguals and Albanian-English Sequential Bilingual Returnees in Tirana, Albania” examines whether native Albanian speakers who returned to the country after spending several years abroad in an English-speaking context show less reliable identification of three Albanian phonological contrasts than monolingual Albanian speakers. Results show that the returnees not only did not perform worse than the monolinguals but actually outperformed them for one contrast that is thought to be undergoing a merger in Albanian. I think that this paper could make a valuable contribution to the literature on L1-L2 interactions in bilingual speakers as well as to research on Albanian phonology/phonetics. The main research question is worth pursuing and the pattern of results is interesting, especially with regard to the possibly-merging contrast. However, I have some concerns about the connection between the theoretical framing of the paper and the literature reviewed, on the one hand, and the experimental work conducted, on the other hand. In addition, there are some terminological issues and theoretical points that I would really like to see clarified. For this reason, I recommend that the paper should be revised and resubmitted before being fully considered for publication in Languages.

Main issues

One very obvious issue with the paper is statistical power, since the group of returnees, which could be considered the experimental group, consists of just 12 individuals. This is clearly not ideal but I understand the difficulties that may arise when recruiting this population. Besides, based on the descriptives and visualizations provided, I do not think this is a major issue for the between-group comparisons performed except perhaps for the rhotic contrast, where more emphasis should probably be made on the fact that the difference between groups was marginally significant. In contrast, the low number of participants is rather problematic for any analysis concerned with individual differences and/or relying on by-participant data, as for example the correlations between age and accuracy with participants split into males and females (with only 12 males) and the correlations concerning participants’ language history. Note that, for example, in the latter, a correlation of r= .68 is almost just about significant. This has to do with the sample size, which indicates that it could be that case that other relatively large correlations may not have reached the alpha criterion because of the reduced number of data points. In short, I think the paper should acknowledge the sample size limitations more and be more cautious when interpreting the correlational results.

Another issue is the conflation of the terms discrimination and identification when referring to the perceptual ability that is tested in the study. The introduction refers to discrimination the whole time but then it turns out that the perception task is an identification/categorization task and not a discrimination task. While I agree that a correct identification of, say, [r] and [ɹ] requires that the two are distinguished or discriminated, using discrimination instead of identification may be confusing for many readers who are familiar with both experimental paradigms. I suggest using “identification” throughout the text. If the author/s wish to keep “discrimination” as is, they should provide a rationale for doing so.

Perhaps the biggest problem I see, and one I really encourage the author/s to address, is that the way in which the present study connects to the previous literature is not as clear as it seems. To begin with, most literature showing L1 attrition effects tested production and not perception. This is not a problem per se but it has to be noted that the affordances of tasks in the two modalities are different. Production is active and there is some optionality to how one may produce specific sounds, whereas perception is passive and the goal is to recognize words (and their sounds) in spite of all the variability. This may have an impact in how likely attrition effects in the two modalities are. Furthermore, previous studies show that attrition is usually observed through minor modifications in the acoustics of specific cues to a contrast (in production) or changes in cue reliance or fairly specific perceptual patterns (perception). This is far from providing evidence of the reduction or loss of a phonological contrast, which I see as the object of study of this paper. Therefore, there is a disconnect between what is reviewed and what is tested in the study and it should be clarified why it is predicted that returnees may actually be worse in perceptual identification of the contrasts examined (is there any evidence that attrition threatens the perceptual identification of L1 phonological contrasts?). This is all the more important considering that no evidence of this is provided in the present study either.

Related to the comment above, an additional problem with interpreting the results is that, if no difference between groups is found, one cannot know if its absence is due to returnees having re-retuned their perception once in Albania or to them never having retuned it at all in the first place. This is briefly mentioned in the paper but I think more emphasis should be made in highlighting this limitation and linking it to prior literature which may speak to the issue. Ideally, this literature should match the modality of the task (i.e., perception) and the ability tested (i.e., perceptual identification). If this literature does not exist, then one must be really cautious when discussing the findings and acknowledge that little can be said about what happened to the returnees’ perception wherever no difference with monolinguals is found.

Finally, I encourage the author/s to emphasize the difference in L1-L2 correspondence that exists between the laterals, on the one hand, and the other two contrasts on the other hand. For the rhotics and the affricate-plosive contrast, you have a scenario in which the L2 has one of the sounds and not the other. For the laterals, you have a contrastive distinction in the L1 surfacing as an allophonic alternation in the L2. If one really expected an influence of L2 usage on L1 speech perception, it would a priori be in the lateral contrast because for it L2 exposure would provide evidence of the contrast NOT being contrastive and, with extensive exposure, this could in turn affect how this contrast is perceived in the L1. This does not happen for the other two contrasts, for which an L2 speaker could well spend years and years in an immersion setting without hearing one of the members of the contrast (i.e., /r/, /c/) in L2 speech. I think this difference is essential to understand why previous research on attrition targeted the laterals and, most importantly, to contextualize the finding that language history variables only seem to reliably relate to accuracy for the laterals.

Minor remarks

Abstract: “Results showed that (1) reaction times for discriminating /c/ versus /t/ were longest, indicating that this contrast was “harder” than the other contrasts”. Mentioning that this was true for both groups would make this clearer.

P. 5, l. 228-234: This paragraph, and particularly the sentence about young female speakers, come a bit as a surprise. It would help if a clearer connection between paragraphs could be worked out.

P. 6, l. 271-273: As mentioned above, it needs to be spelled out why it could be the case that returnees do not distinguish minimal pairs with the target contrasts. Note that here we are not talking about using slightly different cues in perception or having trouble with these contrasts in challenging tasks involving high demands on several levels of processing (e.g., lexical, semantic, syntactic), but just the ability to identify native L1 categories in a task that is not particularly demanding and has a very strong segmental focus (i.e., the design drives listeners attention to the target distinctions). Hence, it is not obvious why this should be difficult for any native speaker of Albanian.

P. 7, l. 308: Here it is stated that the minimum time in an English environment is 1 year but on Table 2 it says that it is 3 years.

P. 10, l. 420: Why did you not include any by-token and by-position random slopes?

P. 15, l. 526-527: here the text first says that RT were slower for rhotics than plosive vs. affricate and immediately afterwards that reaction times were the longest for plosive vs. affricate. Table 8 confirms that it is the latter that show slower RT so the first sentence should be corrected.

P. 16, l. 551-552: I wonder if part of the problem with the model on accuracy may have been that some the language history variables used are correlated and this resulted in multicollinearity. How correlated are ALA, YLA, YAR and %Eng? Using mixed models as for RT would be result in a stronger analysis than the correlations provided, especially in the light of the sample size and the small variance in the scores between participants (see Figure 6), so it would be great if a solution could be found for the overfitting issue.

P. 18, l. 629,-630: A potentially interesting way to explore the hypothesis that L2 experience facilitated identification would be to look at scores for the two sounds separately and check whether the difference between groups is stable across both of them or whether it is larger for the sound that is part of the L2. If the returnees outperformed the monolinguals when the target was /tʃ/ (L2) but not (or not so much) when it was /c/ (not L2), this would be very much in line with the idea that the difference is related to L2 facilitation.

P. 20, l. 727-729: As mentioned above, considering the results there is no way of knowing whether any L1 restructuring took place for the returnees at any point. They were not significantly worse than the monolinguals for any contrast and they outperformed them for the affricate-plosive distinction, which is likely related to this contrast merging. In fact, it would probably be more plausible to argue that those who have restructured their L1 categories across their lifespan are the monolinguals in that they have been losing sensitivity to the distinction, and that this restructuring did not happen to the same degree for the returnees because of their prolonged stays abroad.

Author Response

Thank you for your detailed and helpful comments. We have tried our best to respond to your points and address the concerns that you raised.

Reviewer 2 Report

This study examines the perception of three pairs of phonemic contrasts in Albanian for functional monolinguals and Albanian-English bilingual returnees. The topic and empirical findings are highly relevant and directly contribute to understanding the mechanisms that underlie dynamic bilingual development over the lifespan, here with a particular focus on phonological representations through perception experiments. The paper is clear, well organized, and engages effectively with previous work. In the following, I will offer some minor questions and comments I have on a few points in the paper.     

Abstract: Because the author(s) investigate three sets of binary contrasts, results for all three should be mentioned in the abstract, instead of just /c/ and /tʃ/. 

Ln. 28: Flores (2015, 2020) is some additional work on returnee populations that could be relevant for the author(s) literature review.

P. 2, 1st paragraph: Should the English glosses be in single quotes (e.g., /can/ 'cries')? 

Ln. 167: All sets involve a contrast between a phoneme in English and one not in English. The author(s) discuss how the /l/s are allophonic in English. Could this type of difference (allophones, vs. presence-absence) in the L2 have a potential effect on the maintenance of an L1 contrastive categories? In an overview of issues in L2-development, Eckman et al. (2003:175) discuss this as “allophonic split”.

Ln. 205: I would be useful to see the test phonemes in context of all the consonantal contrasts in Albanian. How do are these contrasts distributed across sets? Are we dealing with different contrastive features/types of contrast? E.g., c/tʃ appears to be manner, l/ɫ appears to be place, and r/ɹ could be either manner and/or place (or maybe something else for all these entirely). How are relevant distinctions distributed across the phoneme inventory? This could potentially influence whether certain features may be activated in English, but maybe for different contrasts. Brown (1998), for example, presents evidence that L1 features may be extended to facilitate L2 perception. Similar L1-L2 mechanisms could be at play here, and in both directions.

Lns. 600–603: In addition to debate on how tight the connection between production and perception may be, there may also be a potential difference between phonological representation (tied to perception) and production (tied to implementation of representation, and subject to motor control constraints and sociolinguistic "feedback"). The present study and work like may contribute to discussions on how access to representations for comprehension and production may be vary based on immersion setting, and contribute to dynamic patterns in production and perception (see Putnam & Sanchez 2013; Putnam et al. 2019 for more detailed proposal).

References

Brown, Cynthia A. The role of the L1 grammar in the L2 acquisition of segmental structure. Second Language Research 14(2): 136–193.

Flores, Cristina. 2015. Losing a language in childhood: a longitudinal case study on language attrition. Journal of Child Language 42(3): 562–590.

Flores, Cristina. 2020. Attrition and Reactivation of a Childhood Language: The Case of Returnee Heritage Speakers. Language Learning 70:S1: 85–121.

Eckman, Fred R. Abdullah Elreyes & Gregory K. Iverson. 2003. Some principles of second language phonology. Second Language Research 19(3): 169–208.

Putnam. Michael T. & Liliana Sánchez. 2013. What’s so incomplete about incomplete acquisition? A prolegomenon to modeling heritage language grammars. Linguistic Approaches to Bilingualism 3(4): 478–508.

Putnam, Michael, Silvia Perez-Cortes & Lilian Sánchez. 2019. Language attrition and the feature reassembly hypothesis. In Monika S. Schmid & Barbara Köpke (eds.), The Oxford Handbook of Language Attrition, 18–24. Oxford: Oxford University Press.

Author Response

Thank you for review. We have tried to address the questions and comments that you raised.

Round 2

Reviewer 1 Report

I have now read the newer version of the manuscript “Sound Change in Albanian Monolinguals and Albanian-English Sequential Bilingual Returnees in Tirana, Albania”. The authors have responded to several of my concerns and I think the manuscript has improved, especially in its theoretical framing. I still think the manuscript has some shortcomings in terms of the experimental evidence provided, but these are now better acknowledged and, besides, as the authors discuss, some of them are very hard to avoid in research of this kind. For these reasons, and because I still think the paper will make a valuable contribution to the study of Albanian in addition to the research devoted to L1 attrition, I recommend the publication of this article. In spite of this, I have some comments that may be helpful to the authors if taken into account. In any case, I leave it to the judgment of the editors to decide whether the points mentioned below should lead to some more revision of the manuscript or not.

1)    I appreciate the paragraphs added to the discussion to emphasize the lack of L2-immersion data and the small sample size but I got the impression that they were written in a style that is slightly defensive and/or confrontational. I do not have a problem with this but I thought I would mention it because it could be that the readers also read it in this way. I really hope this style does not stem from my comments on the previous version of the manuscript. I truly appreciate the work that has gone into this study and I see its value (and therefore recommend publication), but the critical issues with sample size and analyses I brought up had to be mentioned because these are issues for any sort of study relying on quantitative methods, regardless of its subject matter. These issues are largely still there in the present version and the paper may receive some criticism because of them, but I think the acknowledgment (revised or in the current version) is a meaningful addition to the discussion and works well as an exercise of caution.

2)    Note that the authors now mention how perception of /c/ and /tʃ/ differ, reporting that both groups seem to identify /tʃ/ more accurately than /c/, but no statistical analysis backing this up is provided. It would be advisable to include one.

3)    Finally, to me, the issue of the disconnect between this study and previous research still persists to some extent and I want to clarify why, in case it is helpful. Here the study is set up with the expectation that bilinguals, as a group, could be categorically different from monolinguals in their ability to categorically identify L1 phonemes (is this X or Y?), with the assumption that they may have lost sensitivity to the contrasts (pointing towards a perceptual merger). However, it is hard to see how previous research could suggest that this may be the case. De Leew et al. (2018) is now cited several times as providing very strong evidence of L1 attrition in production ­–as a counterpart to this study but in the other modality. However, while De Leew et al. (2018) certainly found some evidence of attrition in individual-level analyses, it is not obvious whether the majority of their participants had really lost sensitivity to the contrast in the way it is operationalized here. Now other references have been added to motivate the paper’s expectations but several also point to the same issue: this assumption and the task used may not be the best way to look for fine-grained perceptual differences pointing to L1 attrition. For example, Gonzales and Lotto (2013) shows different perception in L1 and L2 for bilinguals but they do so with a paradigm that is much more sensitive to gradual differences, and thus more similar to the literature on attrition using production and measuring small differences in fine phonetic detail. In short, my point is, an identification task with naturally produced stimuli (to be entered in the analyses as correct vs. incorrect responses) may not be sensitive enough to complement the findings on production with perception data. For that, looking at perceptual boundaries between phonemes (as in Gonzales and Lotto, 2013) or at the slope of the categorization curves when categorizing speech continua would be more appropriate alternatives and, perhaps, with these more sensitive measures, the results would look very different. I just think that it would be really beneficial to briefly discuss this when reflecting on the results, as I can image that readers with a speech perception background may have similar concerns.

Author Response

Thank you for the additional feedback. We would like to wholeheartedly apologise if some of our previous responses came across as defensive, and much worse, as confrontational. That was certainly not our intention. We very much value the feedback that we received.

Please find attached more detailed responses to the points that you raised.
